# UI2Code$^N$: A Visual Language Model for Test-Time Scalable Interactive UI-to-Code Generation

## Abstract

User interface (UI) programming is a core yet highly complex part of modern software development. Recent advances in visual language models (VLMs) highlight the potential of automatic UI coding, but current approaches face two key limitations: multimodal coding capabilities remain underdeveloped, and single-turn paradigms make little use of iterative visual feedback. We address these challenges with an interactive UI-to-code paradigm that better reflects real-world workflows and raises the upper bound of achievable performance. Under this paradigm, we present UI2Code$^N$, a visual language foundation model trained through staged pretraining, fine-tuning, and reinforcement learning to achieve foundational improvements in multimodal coding. The model unifies three key capabilities: UI-to-code generation, UI editing, and UI polishing. We further explore test-time scaling for interactive generation, enabling systematic use of multi-turn feedback. Experiments on UI-to-code and UI polishing benchmarks show that UI2Code$^N$ establishes a new state of the art among open-source models and achieves performance comparable to leading closed-source models such as Claude-4-Sonnet and GPT-5. Both the model and code will be released.

## 1 Introduction

Recent advances in visual language models (VLMs) have opened up new possibilities for user interface (UI) coding, such as the automatic transformation of UI screenshots into executable code. As user interfaces (UI) are a core component of software systems, automating their development could significantly reduce costs and expand access to front-end application creation. Unlike general programming tasks, UI coding is a cyclical and tightly interwoven process of visual observation, reasoning, and code expression, continually refined through real-time visual feedback. At the same time, UI development poses unique challenges: from grasping overall layouts to correctly identifying nested components, while also capturing subtle visual details such as spacing, color, and typography. Crucially, all of these elements must be faithfully translated into long, executable code.

Although visual language models (VLMs) have made remarkable progress on general vision understanding benchmarks, their performance in UI coding remains notably insufficient. Qualitatively, as shown in Figure 1, even advanced proprietary VLMs such as Gemini-2.5-Pro (Comanici et al., 2025) and Claude-4-Sonnet-Thinking encounter significant challenges in UI-to-code generation. Quantitatively, on the Design2Code benchmark (Si et al., 2024), commercial VLMs like Claude-4-Sonnet achieve only 76.3, falling short of human evaluation standards, while leading open-source VLMs such as Qwen2.5-VL-72B (Bai et al., 2025), InternVL3-78B (Zhu et al., 2025), and Step-3-321B (Team) score below 45/100. The gap becomes even more pronounced on more demanding tasks, such as UI polishing toward target prototypes or instruction-based editing from reference designs, where both open- and closed-source models consistently struggle (Table 1). More recent approaches attempt to orchestrate complex agent-style workflows at inference (Jiang et al., 2025a; Wan et al., 2024; Wu et al., 2025), yet these remain fundamentally constrained by rigid heuristics and the inherent ceiling of current VLM capabilities.

We attribute the current limitations of VLMs in UI coding to two key challenges. First, existing models lack a strong multimodal coding capability, which is essential for reliably translating com-

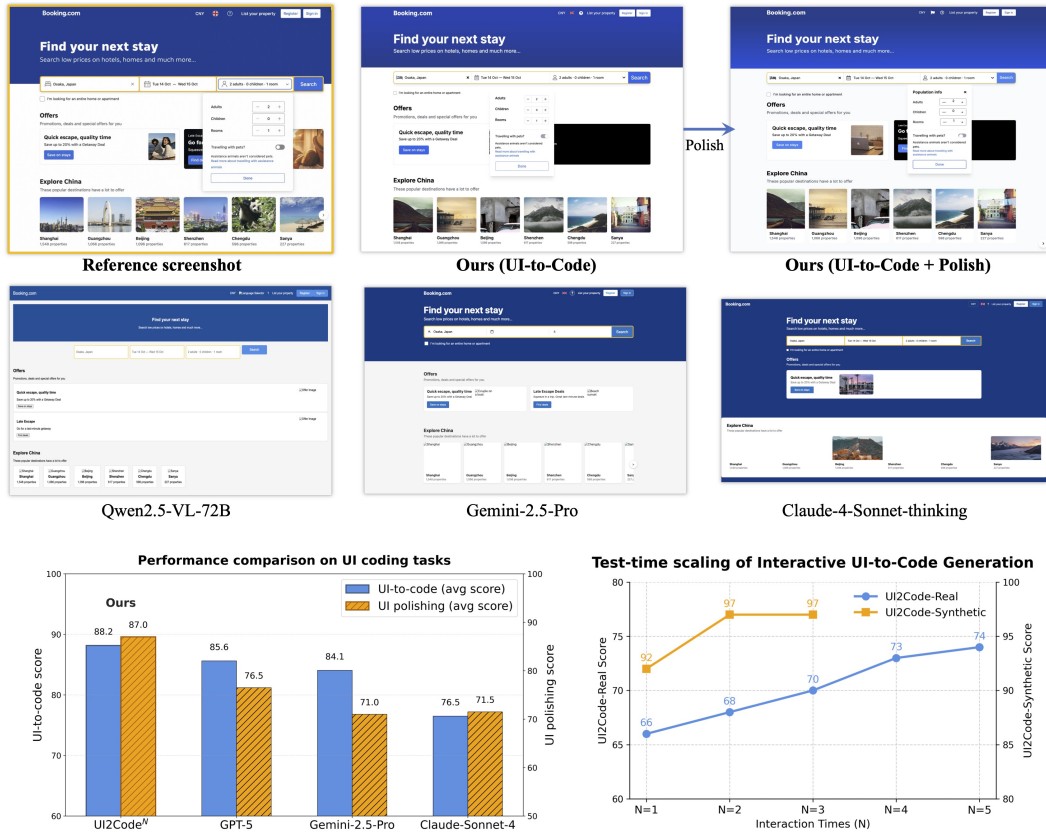

Figure 1: **Top:** Comparison of UI-to-code generation outputs from leading models versus our model, using the same reference screenshot. Our model achieves the highest fidelity, further enhanced by our UI polishing capability. Additional qualitative examples with diverse content, aspect ratios, and layouts are provided in Appendix A.6. **Bottom left:** Performance comparison on UI-to-code and UI polishing tasks. **Bottom right:** Test-time scaling curve of our model on the UI-to-code task, enabled by our interactive UI-to-code paradigm.

plex visual layouts into executable code. This weakness is further compounded by the tension between the complexity of UI-to-code generation, which demands intensive training, and the scarcity of high-quality paired data. Real webpages are abundant but their HTML is noisy and entangled with external resources, whereas synthetic datasets are clean but overly simplistic (Gui et al., 2025; Yun et al., 2024). Second, there is a fundamental disconnect between existing single-turn UI-to-code paradigms and real-world UI development workflows, which limits both their performance ceiling and practical utility. Fundamentally, UI-to-code is inherently an interactive process of *reasoning with visual feedback*: rendered results cannot be inferred from code alone, and runtime factors such as font fallback, browser defaults, and DPI scaling make pixel-level fidelity unverifiable without actual rendering.

In this work, we propose a novel Interactive UI-to-Code paradigm that fundamentally departs from prior single-turn generation approaches, redefining UI-to-code as an iterative and interactive process of generation, editing, and polishing. Such paradigm provides flexible usage with enhanced performance and enables test-time scaling in UI-to-code generation. Guided by this paradigm, we present UI2Code[N], a powerful visual language model trained via a three-stage training pipeline: large-scale pretraining on noisy real-world data to build broad multimodal foundations, supervised fine-tuning on synthetic datasets to improve code quality, and reinforcement learning with a carefully designed verifier to exploit unpaired real webpages while maintaining generation fidelity. Experimental results demonstrate that our UI2Code[N] achieves state-of-the-art performance in UI coding. Building upon the core task of UI-to-code, UI2Code[N] further extends its capabilities to UI polishing and UI editing.

To sum up, our main contributions include:

- We propose Interactive UI-to-Code, a new paradigm that reconceptualizes UI-to-code generation as iterative reasoning and self-correction with visual feedback, enabling flexible code generation and editing. This approach naturally supports test-time scaling, e.g. achieving 12% improvement with four rounds of UI polishing.

- We present UI2Code[N], the first open-source VLM to incorporate UI-to-code, UI polishing and UI editing. UI2Code[N] achieves state-of-the-art results across benchmarks including Design2Code (Si et al., 2024), Flame-React-Eval (Ge et al., 2025) and Web2Code (Yun et al., 2024), surpassing closed-source leading VLMs including Gemini-2.5-Pro and Claude-4-Sonnet, and advancing the open-source UI-to-code frontier by 35% on average.

- This work is the first to present the full training recipe of a foundational coding VLM, systematically covering pre-training, fine-tuning, and reinforcement learning with a novel reward design. Through this three-stage framework, we acquire broad foundational knowledge while balancing data realism and generation quality.

## 2 RELATED WORK

### 2.1 UI-TO-CODE BENCHMARKS

Design2Code (Si et al., 2024) introduced the first benchmark built from real-world webpages, along with visual-centric metrics such as Block-Match and CLIP similarity. Its construction pipeline prunes raw HTML by removing external dependencies and replacing images with placeholders, preserving real-world sources while simplifying the resulting webpages compared to their original distribution. Subsequent benchmarks, including Web2Code (Yun et al., 2024) and Flame-React (Ge et al., 2025), refined the data pipeline but continued to rely heavily on LLM-synthesized HTML. More recently, WebGen-Bench (Lu et al., 2025) broadened the evaluation scope to functional website generation, employing automated agents to test interactivity and functionality.

### 2.2 UI-TO-CODE DATASETS

Progress in UI-to-code generation has been driven largely by dataset scaling. Early large-scale efforts were primarily driven by synthetic data. For instance, WebSight (Laurençon et al., 2024) introduced two million synthetically generated screenshot–code pairs using Tailwind CSS. Similarly, Web2Code (Yun et al., 2024) curated a large-scale instruction-tuning dataset by combining LLM-synthesized data with refined existing resources. Later efforts such as WebCode2M (Gui et al., 2025) and Vision2UI (Gui et al., 2024) constructed million-scale datasets sourced from real-world webpages (e.g., Common Crawl (Common Crawl Foundation, 2007–)), followed by extensive pruning and filtering. While these datasets preserve structural integrity, pruning often leads to oversimplified webpages. Despite their scale, all of the aforementioned datasets either rely on LLM-synthesized content or heavily pruned HTML that removes dependencies such as CSS, thereby limiting their fidelity to complex real-world webpage distributions.

### 2.3 UI-TO-CODE GENERATION MODELS AND SYSTEMS

In contrast, a series of recent works move to leverage commercial VLMs through agent-based workflows. DECLARUI (Zhou et al., 2024) decomposes UI-to-code into detection, segmentation, and classification; DCGen (Wan et al., 2025) adopts a divide-and-conquer strategy; and ScreenCoder (Jiang et al., 2025b) introduces a modular multi-agent framework with grounding, planning, and generation.

## 3 METHOD

### 3.1 INTERACTIVE UI-TO-CODE PARADIGM

UI coding is fundamentally an iterative reasoning problem driven by continuous visual feedback. In real development workflows, developers draft an initial implementation, render it, compare it to the

target design, and correct mismatches. This process forms a closed-loop self-correction mechanism rather than a one-shot mapping. Furthermore, runtime factors such as font fallback or DPI scaling introduce uncertainties that static generation cannot eliminate, making observable feedback essential for achieving reliable visual fidelity.

To capture this, we propose **Interactive UI-to-code generation**, which formulates UI construction as an iterative, feedback-driven transformation process. Let

$$\mathcal{F}_\theta(I, C, R, E) \to C'$$

be a unified transformation function, where $I$ is the target UI image, $C$ is the current code, $R = \text{Render}(C)$ is its rendered output, $E$ denotes optional edit instructions, and $C'$ is the improved code. Existing UI-to-code work corresponds to a restricted, single-step instantiation of this general formulation.

**(1) UI-to-code.** The workflow begins with draft generation:

$$C^{(0)} = \mathcal{F}_\theta^{\text{UI2Code}}(I),$$

which provides a high-quality initial approximation but cannot resolve rendering-dependent discrepancies.

**(2) UI polishing.** Polishing performs iterative refinement through visual feedback. At iteration $t$, the model observes the target image $I$, the current code $C^{(t)}$, and its rendered output $R^{(t)} = \text{Render}(C^{(t)})$, and produces an improved version:

$$C^{(t+1)} = \mathcal{F}_\theta^{\text{Polish}}\left(I, C^{(t)}, R^{(t)}\right).$$

This establishes a self-improvement loop analogous to policy refinement or **test-time scalable optimization**, gradually reducing residual errors. Repeated polishing yields a scalable variant UI2Code$^N$, where larger iteration budgets correspond to higher fidelity.

**(3) UI editing.** Editing adapts existing UIs by conditioning on explicit modification instructions:

$$C' = \mathcal{F}_\theta^{\text{Edit}}(I, C, E).$$

This extends the paradigm from faithful reproduction to interactive, instruction-driven transformation.

Together, these components instantiate a unified, visually grounded transformation framework that integrates drafting, self-correction, and targeted editing. By modeling UI generation as an iterative process with explicit visual feedback, our paradigm overcomes the inherent limitations of one-shot generation and aligns UI-to-code systems with real-world development practice.

## 3.2 MULTI-STAGE TRAINING

Although recent VLMs have demonstrated substantial progress on general vision benchmarks, their performance on UI coding remains limited. The challenges primarily arise from two aspects.

First, *the inherent difficulty of the UI coding task.* The model must accurately perceive UI-style images, capturing fine-grained details such as icons, fonts, and line styles—despite their distributions differing significantly from the natural images used in most pretraining. It must further contend with the complexity of code, as HTML frequently exceeds 10,000 tokens and is densely interwoven with CSS and JavaScript. Beyond these difficulties, precise alignment between UI images and code is required, spanning from global layout structures to individual elements.

Second, *the limitations of available training data.* Although real webpages are abundant, their HTML is often noisy and entangled with external resources, making them unsuitable for direct use. In contrast, synthetic or pruned datasets provide clean structures but lack the richness of real-world complexity (Gui et al., 2025)(Yun et al., 2024). Faced with this trade-off, previous VLMs have typically resorted to synthetic or heavily pruned datasets to ensure basic UI-to-code generation (Laurençon et al., 2024; Yun et al., 2024). However, this reliance excludes large-scale real-world web data from pretraining and shifts complex webpages out of the training domain, thereby constraining performance in practical applications.

To address these challenges, we adopt a three-stage training pipeline. We first conduct continual pre-training on large-scale real-world webpage image–HTML pairs to establish broad UI coding knowledge. We then perform supervised fine-tuning on clean, curated datasets to enhance diverse functionalities such as UI-to-code, UI polishing, and UI editing. Finally, we leverage reinforcement learning to adapt the model to complex real-world distributions without relying on paired ground-truth HTML. Next, we detail each stage together with its tailored data and training strategy.

### 3.2.1 CONTINUAL PRE-TRAINING: FOUNDATIONAL VISION-CODE ALIGNMENT

The primary objective of this stage is to bridge the modality gap between the continuous pixel space of $I$ and the discrete symbolic space of $C$, establishing a robust foundation for understanding the structural complexity of the DOM. We formulate this as an autoregressive next-token prediction task, optimizing the joint probability of code tokens.

To obtain tighter grounding between UI segments and their underlying DOM code, we adopt a GUI-REG–style training objective (Hong et al., 2024). Given the full UI image $I$ and a bounding box $b$ corresponding to a sampled DOM node, the model predicts the corresponding code snippet $C_b$. Formally, we optimize the local alignment loss:

$$\mathcal{L}_{\mathrm{dom}}(\theta) = \mathbb{E}_{(I,C)\sim\mathcal{D},\, b\sim\mathcal{B}(C)} \left[ -\sum_{t=1}^{|C_b|} \log p_\theta(c_{b,t} \mid c_{b,<t}, I, b) \right],$$

where $\mathcal{B}(C)$ is the set of visible bounding boxes in the render tree. This encourages $\mathcal{F}_\theta$ to learn localized visual–code correspondences and control sequence length. Our primary corpus is built by crawling webpages that contain both HTML and full-page screenshots, yielding $\sim$10M UI–code pairs. Direct use of *Common Crawl* (Common Crawl Foundation, 2007–) proved infeasible due to missing components (e.g., figures, CSS) that hinder faithful rendering. Instead, we use its URLs as seeds for large-scale crawling, followed by tag whitelisting and redundancy removal.

To diversify data distributions and ensure global coherence, we incorporate high-fidelity UI–code datasets such as WebCode2M (Gui et al., 2025) and WebSight Laurençon et al. (2024). While their HTML may be synthetic, they preserve the crucial alignment between $I$ and the rendered output $R$. These provide informative supervision for whole-page consistency via the standard pair loss:

$$\mathcal{L}_{\mathrm{pair}}(\theta) = \mathbb{E}_{(I,C)\sim\mathcal{D}_{\mathrm{pair}}} \left[ -\sum_{t=1}^{|C|} \log p_\theta(c_t \mid c_{<t}, I) \right].$$

To preserve general VLM capabilities, we interleave coding-related samples with broad tasks (captioning, VQA, OCR, grounding) and over 1B tokens of language–code data. The final continual pre-training objective aggregates these multi-source constraints via a weighted average proportional to the sampling ratios of their respective datasets. Training is initialized from an early checkpoint of GLM-4.1V-9B-Base (Hong et al., 2025), with a learning rate of 2e-5, tensor parallel size of 2, and a global batch size of 1,536. Continual pre-training covers $\sim$20M vision–code samples in total.

### 3.2.2 SUPERVISED FINE-TUNING: DIVERSE CAPABILITY ALIGNMENT

In the SFT stage, we instantiate the unified transformation function $\mathcal{F}_\theta(I, C, R, E)$ to support complex interaction and reasoning. To facilitate the transition toward the deep thinking paradigm and enhance problem-solving depth, we explicitly structure the model's output into a thinking process block and a final answer block: `<think>`$\{T\}$`</think><answer>`$\{C'\}$`</answer>`. This design serves two critical purposes: first, it forces the model to manifest a latent reasoning trajectory $T$—such as analyzing layout constraints or diagnosing visual bugs—before generating code, thereby improving capability on complex tasks. Second, it standardizes the output format for the subsequent RL stage, allowing the environment to reliably extract the final code $C'$ for rendering and verification while isolating the thought process.

The optimization objective is the autoregressive likelihood of the thought-augmented sequence conditioned on task-specific inputs $\mathcal{X}$:

$$\mathcal{L}_{\mathrm{SFT}}(\theta) = \mathbb{E}_{(\mathcal{X},T,C')\sim\mathcal{D}_{\mathrm{SFT}}} \left[ -\log p_\theta(T, C' \mid \mathcal{X}) \right].$$

Depending on the sub-task, the input $\mathcal{X}$ takes different forms:

- **UI-to-Code** ($\mathcal{X} = \{I\}$)**:** The model plans the global layout structure in $T$ before generating $C^{(0)}$.
- **UI Polishing** ($\mathcal{X} = \{I, C, R\}$)**:** The thinking process $T$ acts as a *visual debugger*, explicitly comparing the target $I$ against the current render $R$ to identify discrepancies (e.g., alignment shifts, style errors) before outputting the corrected $C'$.
- **UI Editing** ($\mathcal{X} = \{I, C, E\}$)**:** The model interprets the edit instruction $E$ to locate the modification site within $C$.

To ensure data fidelity—a major bottleneck in UI generation—we employ a "reverse-engineering" strategy using state-of-the-art models. We first generate complex ground-truth HTMLs, then synthetically derive the "buggy" inputs (for polishing) or instructions (for editing). This ensures that the thinking process $T$ implies a valid causal path to the code correction. The detailed curation process is illustrated in Appendix A.3 In total, we construct 80K high-quality samples, and train for 5 epochs with sequence length of 32,768, batch size of 256 (with packing) and learning rate of 5e-6.

### 3.2.3 REINFORCEMENT LEARNING

RL enables optimizing directly for visual alignment rather than token-level likelihood, bridging the gap between generation and human perception, and avoiding noisy HTML ground truth. We refine the policy $\pi_\theta$ using Group Relative Policy Optimization (GRPO) (Shao et al., 2024). For each input $x$, we sample a group of outputs $\{y_1, \ldots, y_G\}$ from the old policy $\pi_{\theta_{\text{old}}}$ and optimize the following objective:

$$\mathcal{J}(\theta) = \mathbb{E} \left[ \frac{1}{G} \sum_{i=1}^{G} \min \left( \rho_i A_i, \, \text{clip}(\rho_i, 1 - \epsilon, 1 + \epsilon) A_i \right) \right], \tag{1}$$

where $\rho_i$ is the importance ratio. The advantage $A_i$ is derived from the group-normalized reward scores: $A_i = (r_i - \bar{r})/\sigma_r$. Crucially, we eschew KL regularization to prevent the policy from being overly constrained by the SFT model, thereby raising the performance ceiling.

**Reward Design** The effectiveness of GRPO hinges on the reward signal $r_i$. We implement a hierarchical reward structure to balance strict instruction following with visual fidelity:

- **Format Penalty:** Responses failing syntax constraints receive a hard penalty ($r_i = -1$).
- **Quality Reward:** Valid responses are routed to a dense visual assessment. Since standard metrics often fail to capture UI nuances, designing a robust visual reward $r_{\text{quality}}$ presents unique challenges regarding calibration and stability.

During experiments, we find that quality Reward design is a pivotal challenge in UI generation. Standard metrics like CLIP are brittle: they are overly sensitive to global layout shifts yet blind to local UI nuances. Consequently, we adopt visual language models (specifically GLM-4.5V) for scalable evaluation. Yet, a key difficulty persists: *calibration instability*, characterized by inconsistent scoring of visually similar candidates. We tackle this through three progressive refinements:

- **Scoring via Verifier (Alg. 1):** As a baseline, we compute an absolute score $S = \texttt{verifier\_score}(I_{\text{target}}, I_{\text{cand}})$. While straightforward, independent queries suffer from severe calibration drift across different samples.
- **Scoring via Comparator (Alg. 2):** To mitigate drift, we introduce a relative comparator $\texttt{comp\_score}$ that evaluates the candidate against the previous step's reference in a single query. This anchors the score locally but ignores relationships between concurrent candidates.
- **Comparator + Round-Robin (Alg. 3):** To ensure global fairness across the candidate pool ($N \sim 16$), we adopt a tournament-style approach. Candidates are compared pairwise, and the reward is derived from the total win count, yielding the most robust ranking.

Our ablation studies confirm that Alg. 3, despite higher computational cost ($O(N^2)$), significantly outperforms the others in alignment accuracy and is therefore adopted as our primary method. Further implementation details are provided in Appendix A.1.

**Algorithm 1** Scoring via Verifier

**Req.** $I_{\text{target}}$, $I_{t-1,\text{ref}}$, $\{I_{t,i}\}_{i=1}^{N}$
**Out.** $\text{Reward}[t,i]$
1: $S_{\text{ref}} \leftarrow \text{verifier\_score}(I_{\text{target}}, I_{t-1,\text{ref}})$
2: **for** $i = 1, \ldots, N$ **do**
3: $\quad S_i \leftarrow \text{verifier\_score}(I_{\text{target}}, I_{t,i})$;
$$\text{Reward}[t,i] \leftarrow \begin{cases} -1, & I_{t,i} \text{ fails to render,} \\ 0, & S_i \leq S_{\text{ref}}, \\ S_i & S_i > S_{\text{ref}}. \end{cases}$$
4: **end for**

**Algorithm 2** Scoring via Comparator

**Req.** $I_{\text{target}}$, $I_{t-1,\text{ref}}$, $\{I_{t,i}\}_{i=1}^{N}$
**Out.** $\text{Reward}[t,i]$
1: **for** $i = 1, \ldots, N$ **do**
2: $\quad S_{\text{ref}}, S_i \leftarrow \text{comp\_score}(I_{\text{target}}, I_{t-1,\text{ref}}, I_{t,i})$;
$$\text{Reward}[t,i] \leftarrow \begin{cases} -1, & I_{t,i} \text{ fails to render,} \\ S_i & S_i \leq S_{\text{ref}}, \\ S_i & S_i > S_{\text{ref}}. \end{cases}$$
3: **end for**

**Algorithm 3** Scoring via Comparator + Round-Robin

**Req.** $I_{\text{target}}$, $I_{t-1,\text{ref}}$, $\{I_{t,i}\}_{i=1}^{N}$
**Out.** $\text{Reward}[t,i]$
1: Define $\text{Pool} = \left\{ i \,\Big|\, I_{t,i} \text{ renders and } (S_{\text{ref}}, S_i) = \text{comp\_score}(I_{\text{target}}, I_{t-1,\text{ref}}, I_{t,i}),\ S_i > S_{\text{ref}} \right\}$.
2: **for** $i = 1, \ldots, N$ **do**
3: $\quad \text{Reward}[t,i] \leftarrow \begin{cases} -1, & \text{if } I_{t,i} \text{ fails to render,} \\ 0, & \text{if } i \notin \text{Pool,} \\ 1 + \sum\limits_{\substack{j \in \text{Pool} \\ j \neq i}} \left( \mathbf{1}[S_i > S_j] + \frac{1}{2}\mathbf{1}[S_i = S_j] \right), & \text{if } i \in \text{Pool,} \end{cases}$
$\quad$ where $(S_i, S_j) = \text{comp\_score}(I_{\text{target}}, I_{t,i}, I_{t,j})$ for pairwise comparisons.
4: **end for**

**Implementation**  We train on a mixture of 12K real-world (Mind2Web) and 30K synthetic examples. To enhance robustness, input prompts are diversified using GLM-4.5V, Claude-3.5-Sonnet, and iterative self-correction (UI2Code$^N$, $N \sim \mathcal{U}[1,4]$). We employ a batch size of 64 with a group size of $G = 16$, training for 400 steps.

## 4 Experiments

### 4.1 Evaluation Setup

**Benchmarks:**  To evaluate the effectiveness of UI2Code$^N$ on UI-to-Code generation task, we conduct experiments on several widely used benchmarks, including Design2Code (Si et al., 2024), Flame-React-Eval (Ge et al., 2025), and Web2Code (Yun et al., 2024). However, these benchmarks primarily consist of relatively simple screenshots that may not fully capture the complexity of real-world webpages. To address this limitation, we further construct *UI2Code-Real*, a benchmark of 115 webpages collected from in-the-wild sources. This benchmark serves as a more realistic evaluation setting, allowing us to assess whether models trained with synthetic and curated data can generalize effectively to real-world UI-to-code scenarios. For the UI polishing task, we further construct *UIPolish-bench*, which consists of 100 synthetic webpages and 100 real-world webpages, providing a balanced evaluation of both controlled and in-the-wild scenarios. A more detailed description of these benchmarks, along with our curated *UIPolish-bench* and *UI2Code-Real*, is provided in Appendix A.5.

**Evaluation Metrics:**  We consider two main evaluation approaches: (1) **CLIP scoring**, which uses CLIP-based similarity to assess semantic alignment, as in Design2Code (Si et al., 2024); and (2) **VLM scoring**, which leverages visual large language models (VLMs) to provide human-aligned judgments of design fidelity and usability, as in Web2Code (Yun et al., 2024). In this work, motivated by the stronger visual and semantic capabilities of VLMs, we follow Hong et al. (2025) and adopt VLM-based scoring metrics. This choice is further supported by our reinforcement learning ablation studies (Sec. 4.3.2), where VLM rewards consistently outperform CLIP-based ones. For UI

Table 1: Experimental results on UI-to-Code and UI Polishing benchmarks. **Bold** text indicates the best score among open-source models, and underlined text indicates the best score across all models.

| Model | UI-to-Code | | | | UI Polishing | |
| --- | --- | --- | --- | --- | --- | --- |
| | Design2Code | Flame | Web2Code | UI2Code-Real | UIPolish-Real | UIPolish-Synthetic |
| **Open-source VLM** | | | | | | |
| InternVL3-9B | 15.3 | 11.3 | 12.3 | 16.5 | 4.0 | 7.0 |
| InternVL3-78B | 30.0 | 51.3 | 45.5 | 30.4 | 10.0 | 15.0 |
| Qwen2.5-VL-7B | 29.1 | 25.0 | 37.2 | 26.1 | 11.0 | 14.0 |
| Qwen2.5-VL-72B | 41.9 | 46.3 | 64.1 | 40.9 | 23.0 | 38.0 |
| MiMo-VL-7B-SFT | 28.3 | 10.0 | 44.3 | 33.9 | 17.0 | 33.0 |
| MiMo-VL-7B-RL | 28.7 | 8.8 | 38.3 | 30.4 | 16.0 | 30.0 |
| Kimi-VL-A3B-Instruct | 27.3 | 50.0 | 69.1 | 26.1 | 14.0 | 40.0 |
| Kimi-VL-A3B-Thinking | 38.8 | 36.3 | 46.6 | 27.0 | 14.0 | 27.0 |
| GLM-4.1V-9B-Thinking | 64.7 | 72.5 | 71.3 | 53.0 | 42.0 | 46.0 |
| **Closed-source VLM** | | | | | | |
| Claude-4-Sonnet-thinking | 81.2 | 76.3 | 85.1 | 63.5 | 78.0 | 65.0 |
| Claude-3.7-Sonnet-thinking | 77.7 | 80.0 | 73.3 | 55.8 | 75.0 | 62.0 |
| GPT-5 | 89.7 | 91.3 | 93.7 | 67.8 | 85.0 | 68.0 |
| GPT-4o | 35.3 | 75.0 | 62.7 | 21.7 | 26.0 | 14.0 |
| o4-mini | 63.8 | 83.8 | 77.9 | 59.1 | 65.0 | 65.0 |
| Gemini-2.5-pro | 89.5 | 87.5 | 90.6 | 68.7 | 74.0 | 68.0 |
| Gemini-2.5-flash | 70.5 | 72.5 | 85.7 | 62.6 | 17.0 | 24.0 |
| Doubao-1.5-thinking-vision | 53.7 | 78.8 | 55.6 | 38.3 | 51.0 | 61.0 |
| Doubao-1.6-thinking-250715 | 62.4 | 67.7 | 67.2 | 43.4 | 61.0 | 67.0 |
| UI2Code$^N$-9B-SFT | 79.3 | 85.0 | 80.8 | 67.0 | 76.0 | 89.0 |
| UI2Code$^N$-9B-RL | **88.6** | **95.0** | **92.5** | **76.5** | **80.0** | **94.0** |

polishing, we design an evaluation protocol based on comparison with the original UI screenshot. Given an initial screenshot $A$, the model first generates a corresponding rendering $B$ through UI-to-code generation, followed by a polished rendering $C$. The evaluation then compares whether $C$ is visually closer to $A$ than $B$. If $C > B$ in similarity to the ground-truth design, we count the instance as a successful polish and increment the accuracy by one.

## 4.2 MAIN RESULTS

To verify the effectiveness of UI2Code$^N$, we carried out experiments on two types of UI coding tasks, including UI-2 code generation and UI polishing. Table 1 reports the experimental results compared with both open-source and closed-source VLMs. Compared to several open-source VLMs, our proposed UI2Code$^N$-9B-SFT and UI2Code$^N$-9B-RL achieve substantial improvements across all benchmarks. In particular, on the UI-to-code benchmarks (a three public benchmarks like Design2Code, Flame, Web2Code, and a curated real-world UI2Code-Real benchmark), UI2Code$^N$ demonstrates consistent and significant gains. Notably, the performance of open-source VLMs on UI polishing is generally unsatisfactory. As shown in Table 1, all open-source VLMs achieve less than 50% accuracy on both real and synthetic polishing benchmarks. Intuitively, we set 50% as a threshold: if the probability of successfully polishing a given UI screenshot falls below 50%, the model effectively fails to demonstrate a reliable polishing capability. Under this criterion, existing open-source VLMs cannot be regarded as possessing genuine UI polishing ability. In contrast, our UI2Code$^N$-9B-RL achieves 80.0% on UIPolish-Real and 94.0% on UIPolish-Synthetic, surpassing all open-source models by a large margin and even matching the performance of leading closed-source systems such as Claude-4-Sonnet-thinking and Gemini-2.5-pro. These results verify that our interactive paradigm, coupled with multi-stage training, not only strengthens UI-to-code generation but also equips the model with robust UI polishing capability.

**Test-Time Scaling with UI Polishing.** The interactive UI-to-code generation paradigm endows UI2Code$^N$ with the ability to perform test-time scaling. Specifically, for a UI-to-code generation

task, we begin with an initial round of generation and then recursively refine the output by polishing the UI using the HTML and renderings produced in the previous round. To assess this approach across diverse web pages, we conduct experiments on both real and synthetic subsets of our self-constructed UI2Code benchmark, evaluating performance over multiple interaction rounds $N$, where $N = 1$ corresponds to a single round of generation without polishing. The results in Table 2 show that performance steadily improves on both real and synthetic datasets as the number of interaction rounds increases. Interestingly, performance on UI2Code-Synthetic saturates early at $N = 3$, likely due to the lower difficulty of synthetic data (thus we omit evaluations for $N > 3$). In contrast, performance on UI2Code-Real continues to improve consistently from $N = 1$ through $N = 5$.

Table 2: Test-time scaling performance of interactive UI-to-code generation

| Benchmark | N = 1 | N = 2 | N = 3 | N = 4 | N = 5 |
|---|---|---|---|---|---|
| UI2Code-Real | 66.0 | 68.0 | 70.0 | 73.0 | 74.0 |
| UI2Code-Synthetic | 92.0 | 97.0 | 97.0 | – | – |

### 4.3 ABLATION STUDY: THE IMPACT OF REWARD DESIGN

In this section, we conduct a thorough ablation study on the impact of RL reward design, including both UI polishing and UI-to-code. For all ablation experiments, we start from the SFT checkpoint of UI2Code$^N$, and run RL with a batch size of 32, rollout number of 16, and learning rate of 1e-6.

#### 4.3.1 REWARD DESIGN FOR UI POLISHING

For UI polishing, we design the reward functions described in Section 3.2.3. We evaluate three strategies for assessing UI polishing performance: the vanilla verifier, the verifier with a comparator function, and the verifier with both a comparator function and a round-robin strategy. The results in Table 3a highlight two key findings regarding reward design. First, the round-robin comparator verifier consistently achieves the best results. Unlike reward designs based on local judgments, this approach enables global ranking across candidates, aligning more closely with the practical goal of UI refinement—selecting the best improvement among alternatives. Second, the effectiveness of comparator-based rewards depends heavily on the reliability of the underlying vision-language model (VLM). Using GLM-4.5V without fine-tuning as the verifier reduced accuracy and degraded performance by 3%, whereas our tailored verifier produced substantial gains. This underscores that weak or noisy reward signals can misdirect reinforcement learning rather than improve it.

Table 3: Comparison of different reward designs across tasks. (a) UI polishing. (b) UI-to-code.

| RL Reward Design | UIPolish-Synthetic | UIPolish-Real | Avg. |
|---|---|---|---|
| SFT | 89.0 | 76.0 | 82.5 |
| + RL (vanilla verifier) | 91.0 | 75.0 | 83.0 |
| + comparator verifier | 93.0 | 78.0 | 85.5 |
| + round-robin strategy | 93.0 | 79.0 | 86.0 |

| Method | Design2Code | Flame |
|---|---|---|
| UI2Code$^N$-SFT | 72.3 | 85.0 |
| RL with CLIP Reward | 62.0 | 72.2 |
| RL with GLM-4.5V Reward | 74.6 | 89.0 |

(a) Reward design in UI polishing task

(b) Reward design in UI-to-code generation task

#### 4.3.2 REWARD DESIGN FOR UI-TO-CODE GENERATION

To enable reinforcement learning for UI-to-code generation, we leverage automatic similarity measures and human-aligned judgments to investigate their effectiveness as reward signals. Specifically, we design two experimental settings: 1) the **CLIP score** (Radford et al., 2021) is employed to provide a continuous and fine-grained reward that reflects semantic consistency between the rendered UI from the generated HTML code and the original UI screenshot; and 2) the **VLM score**, where open-source visual language models (e.g., GLM-4.5V) offer human-aligned evaluations of layout fidelity. As shown in Table 3 (b), GLM-4.5V reward consistently surpasses CLIP reward across both the Design2Code and Flame-React-Eval benchmarks. Moreover, we observe that using a CLIP reward fails to improve performance and in fact leads to degradation compared to the SFT baseline. This indicates that purely visual similarity signals are insufficient for capturing the semantic and

structural fidelity required in UI-to-code generation, and may even misguide the optimization process. These findings highlight the importance of reward design: relying solely on visual similarity metrics may misalign reinforcement learning, whereas VLM-based rewards provide richer and more reliable feedback.

### 4.4 ABLATION STUDY: THE IMPACT OF REAL-WORLD WEBPAGES IN RL STAGE

To further investigate the role of real-world webpages in the reinforcement learning (RL) stage of UI2Code$^N$, we conduct a controlled comparison under identical data budgets (20k RL samples) and training steps (100 iterations) to isolate the effect of incorporating real webpages in the RL stage. While synthetic datasets provide controlled environments with clean labels and diverse coverage of UI patterns, they may fail to capture the complexity, noise, and distributional shifts that occur in real-world interfaces. To bridge this gap, we augment the RL stage with a curated set of real webpages, where the original UI screenshots are collected from in-the-wild sources.

Table 4 demonstrates that including real webpages in the RL stage consistently improves both semantic fidelity and rendering quality. Notably, the improvement is more pronounced on evaluation benchmarks that share similar distributional characteristics with real-world webpages, highlighting the necessity of real data for bridging the sim-to-real gap in UI-to-code generation.

Table 4: The impact of real-world webpage data in reinforcement learning stage.

| RL Data | Design2Code | UI2Code-Real | UIPolish-Synthetic | UIPolish-Real |
|---|---|---|---|---|
| without real data | 81.5 | 68.7 | 92.0 | 65.0 |
| with real data | 82.4 | 75.0 | 93.0 | 80.0 |

## 5 CONCLUSION

We introduce UI2Code$^N$, a 9B-parameter vision–language model with advanced UI coding capabilities. We further propose the Interactive UI-to-Code Generation paradigm, which formulates UI-to-code as an iterative, interactive process that extends both performance and applicability. UI2Code$^N$ achieves state-of-the-art results on UI-to-code and UI polishing benchmarks, surpassing leading VLMs including Claude-4-Sonnet, and Gemini-2.5-Pro. We open-source UI2Code$^N$ to support broader adoption and research.

## ETHICS STATEMENT

This work investigates UI-to-code modeling and involves constructing a large corpus of webpage screenshots and corresponding HTML/CSS code solely for training purposes. We summarize our data governance, privacy protection, and licensing considerations below.

**Data Collection.** URL seeds from the publicly available Common Crawl index are used only to locate webpages. The webpage content used for training is collected independently by our crawler, which strictly respects `robots.txt`, domain-level crawling policies, and rate limits. We do not access login-protected, paywalled, or user-specific content.

**Privacy and PII Protection.** To protect user privacy, we apply automated and rule-based filtering procedures to remove pages containing personally identifiable information (PII), such as names, emails, phone numbers, session-dependent content, or user account information. Pages that include sensitive or identifiable user data are discarded before training.

**Copyright and Licensing.** Webpages may contain copyrighted material owned by their respective authors. To mitigate copyright-related risks, we exclude domains whose Terms of Service disallow automated crawling or derivative processing, avoid collecting or redistributing images or other protected assets, and do not release any raw webpage content (including screenshots or source code). Only the trained model is released, which captures general structural and stylistic patterns rather

than verbatim webpage content. Our model is trained on top of an Apache-2.0 licensed base model and is released under a research-only, non-commercial license.

**Intended Use.** The model is released exclusively for research purposes to support reproducibility and future development in UI-to-code modeling. It is not intended for applications involving sensitive personal data, high-stakes decision making, or commercial deployment without further licensing review.

We believe these measures align with community standards for responsible data governance and ethical AI development.

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

# A APPENDIX

## A.1 REWARD DESIGN

Here we illustrate our reward design in detail. We progressively refine our reward design. Each refinement builds on the previous one:

- **Algo. 4** establishes a baseline verifier scoring method;
- **Algo. 5** corrects calibration drift between candidates and references;
- **Algo. 6** ensures fairness across all candidates through pairwise round-robin comparisons.

This sequence yields steady improvements in RL performance, with the final design achieving the strongest results. Here we denote the target UI as $I_{\text{target}}$, the current UI-polish round number as $t$, and the reference webpage image that the last round generated as $I_{t-1,\text{ref}}$.

**Reward Algo. 4: Verifier scoring.** In the baseline design, we define `verifier_score(target, candidate)` $\in [0,1]$ as the similarity score returned by the VLM when comparing a candidate image against the target. Each candidate $I_{t,i}$ is independently scored as $S_i = $ `verifier_score`$(I_{\text{target}}, I_{t,i})$. We also compute the reference score $S_{\text{ref}} = $ `verifier_score`$(I_{\text{target}}, I_{t-1,\text{ref}})$. Rewards are then defined as $S_i$, with penalties of $-1$ for failed rendering and $0$ for candidates worse than the reference. While simple, this approach suffers from calibration drift, since $S_i$ and $S_{\text{ref}}$ come from separate queries.

**Reward Algo. 5: Comparator scoring.** To place $S_i$ and $S_{\text{ref}}$ under the same calibration, we introduce a pairwise comparator `comp_score(target, cand1, cand2)`. Given a target, it evaluates both the candidate $I_{t,i}$ and the reference $I_{t-1,\text{ref}}$ within a single query, returning two scores that are directly comparable. This eliminates scale inconsistency between $S_i$ and $S_{\text{ref}}$ observed in Algo. 4. Since off-the-shelf VLMs remain unreliable in multi-image comparison, we fine-tune GLM-4.5V with SFT to improve accuracy and robustness.

**Reward Algo. 6: Comparator with round-robin.** Beyond calibration, we also address fairness among multiple rollouts $\{I_{t,i}\}_{i=1}^{N}$. Evaluating them separately still risks misaligned similarity scores across queries. A naive solution would input all $N$ rollouts jointly, but this is impractical since $N$ is large ($\sim 16$ in our setup) and current VLMs degrade with many images. Instead, we adopt a round-robin scheme: candidates are compared pairwise under the comparator, and each $I_{t,i}$ is assigned a score equal to its number of wins. This design achieves consistent and fair ranking across all candidates, at the cost of $O(N^2)$ verifier calls.

---

**Algorithm 4** Scoring via Verifier — Round $t$

---

**Require** target image $I_{\text{target}}$, reference image from round $(t-1)$ denoted $I_{t-1,\text{ref}}$, candidate set $\{I_{t,1}, \ldots, I_{t,N}\}$
**Output** reward map $\text{Reward}[t, i]$
1: Initialize $\text{Reward}[t, i] \leftarrow 0$ for all $i$
2: **for** $i \leftarrow 1$ **to** $N$ **do**
3:      attempt to render $I_{t,i}$
4:      **if** rendering fails **then**
5:          $\text{Reward}[t, i] \leftarrow -1$
6:          **continue**
7:      **end if**
8:      $S_{\text{ref}} \leftarrow \text{VLM\_verifier\_score}(I_{\text{target}}, I_{t-1,\text{ref}})$
9:      $S_i \leftarrow \text{VLM\_verifier\_score}(I_{\text{target}}, I_{t,i})$
10:     **if** $S_i \leq S_{\text{ref}}$ **then**
11:         $\text{Reward}[t, i] \leftarrow 0$
12:     **else**
13:         $\text{Reward}[t, i] \leftarrow S_i$
14:     **end if**
15: **end for**
16: **return** Reward

---

---

**Algorithm 5** Scoring via Comparator — Round $t$

---

**Require** target image $I_{\text{target}}$, reference image from round $t-1$ denoted $I_{t-1,\text{ref}}$, candidate set $\{I_{t,1}, \ldots, I_{t,N}\}$
**Output** reward map $\text{Reward}[t, i]$
 1: Initialize $\text{Reward}[t, i] \leftarrow 0$ for all $i$
 2: **for** $i \leftarrow 1$ **to** $N$ **do**
 3:     attempt to render $I_{t,i}$
 4:     **if** rendering fails **then**
 5:         $\text{Reward}[t, i] \leftarrow -1$
 6:         **continue**
 7:     **end if**
 8:     $(S_{\text{ref}}, S_i) \leftarrow \text{VLM\_comparator\_score}(I_{\text{target}}, I_{t-1,\text{ref}}, I_{t,i})$
 9:     **if** $S_i \leq S_{\text{ref}}$ **then**
10:         $\text{Reward}[t, i] \leftarrow 0$
11:     **else**
12:         $\text{Reward}[t, i] \leftarrow S_i$
13:     **end if**
14: **end for**
15: **return** Reward

---

**Algorithm 6** Scoring via Comparator and Round-Robin — Round $t$

---

**Require** target image $I_{\text{target}}$, reference image from round $t$ denoted $I_{t-1,\text{ref}}$, candidate set $\{I_{t,1}, \ldots, I_{t,N}\}$
**Output** reward map $\text{Reward}[t, i]$
 1: Initialize $\text{Reward}[t, i] \leftarrow 1$ for all $i \in \{1, \ldots, N\}$
 2: $\text{Pool} \leftarrow \emptyset$                                                 ▷ candidates that pass the first-stage screening
 3: **First-stage screening (vs. round-$t-1$ reference)**
 4: **for** $i \leftarrow 1$ **to** $N$ **do**
 5:     attempt to render $I_{t,i}$
 6:     **if** rendering fails **then**
 7:         $\text{Reward}[t, i] \leftarrow -1$                                     ▷ hard penalty
 8:         **continue**
 9:     **end if**
10:     $(S_{\text{ref}}, S_i) \leftarrow \text{VLM\_comparator\_score}(I_{\text{target}}, I_{t-1,\text{ref}}, I_{t,i})$
11:     **if** $S_i \leq S_{\text{ref}}$ **then**
12:         $\text{Reward}[t, i] \leftarrow 0$                                 ▷ no improvement over round-$t-1$
13:     **else**
14:         $\text{Pool} \leftarrow \text{Pool} \cup \{i\}$                                 ▷ kept for round-robin
15:     **end if**
16: **end for**

17: **Round-robin among candidates of round $t$**
18: **for all** unordered pairs $\{i, j\} \subseteq \text{Pool}$ with $i \neq j$ **do**
19:     $(S_i, S_j) \leftarrow \text{VLM\_comparator\_score}(I_{\text{target}}, I_{t,i}, I_{t,j})$
20:     **if** $S_i > S_j$ **then**
21:         $\text{Reward}[t, i] \leftarrow \text{Reward}[t, i] + 1$
22:     **else if** $S_j > S_i$ **then**
23:         $\text{Reward}[t, j] \leftarrow \text{Reward}[t, j] + 1$
24:     **else**
25:         $\text{Reward}[t, i] \leftarrow \text{Reward}[t, i] + 0.5$
26:         $\text{Reward}[t, j] \leftarrow \text{Reward}[t, j] + 0.5$
27:     **end if**
28: **end for**
29: **return** Reward

---

## A.2 REWARD IMPLEMENTATION

### A.2.1 GLM-4.5V VISUAL SCORE FOR UI-TO-CODE RL TRAINING

To provide a reward signal for the RL stage of UI-to-code training, we employ GLM-4.5V as a visual evaluator. Given the original UI screenshot and the rendering generated from the rollout HTML code during training, GLM-4.5V produces a similarity score for the rendered output. Specifically, it assigns a value in the range 0–100, reflecting how closely the rendering matches the ground-truth

screenshot. We then normalize this score to the range $[0, 1]$ and use it as the reward signal. This continuous formulation provides fine-grained feedback, enabling more stable optimization compared to binary success/failure rewards.

The prompt provided to the GLM-4.5V model for this evaluation is as follows:

---

**Prompt**

You will be given two images:

The first image is the reference image (design draft or target rendering).

The second image is the code rendering, which is generated based on the first image using HTML/CSS/frontend code.

Your task is as follows:

Compare the overall similarity between the two images, on a scale from 0 to 100:

- 0 means completely dissimilar.

- 100 means perfectly identical.

When scoring, you should comprehensively consider the following aspects:

- Layout (whether the structural positions are consistent)

- Color scheme (whether the colors are faithfully reproduced)

- Typography (font, font size, line spacing, etc.)

- Spacing and alignment (whether element spacing and alignment are accurate)

- Fine details (button styles, icons, shadows, borders, etc.)

Strictly follow the output format below:

First, provide the final score, where the value **must** be enclosed in LaTeX `\\boxed{}`. Then, provide a justification for the score, explaining which aspects are similar, which aspects differ, and the main factors influencing the score.

---

### A.2.2 OUR VERIFIER VISUAL SCORE FOR UI POLISHING RL TRAINING

To provide a reliable reward signal for reinforcement learning in the UI polishing task, we design a visual verifier that evaluates the fidelity of polished renderings against the original UI screenshots. Given an initial rendering $B$ and its polished counterpart $C$, the verifier compares both with the ground-truth screenshot $A$ and produces a similarity score. Specifically, the verifier assigns a score in the range of 0–100 based on multiple visual dimensions such as layout, color, typography, spacing, and fine-grained details. This raw score is then normalized to the range $[0, 1]$ and used as a reward signal for training. In our RL framework for UI polishing, we adopt a **triplet-based evaluation scheme**. Given the reference screenshot $A$, the initial rendering $B$, and the polished rendering $C$, the visual verifier computes similarity scores $\text{score}(A, B)$ and $\text{score}(A, C)$. The reward is then defined as the normalized score of the polished output:

$$r = \frac{\text{score}(A, C)}{100}, \quad r \in [0, 1]. \tag{2}$$

In addition to this absolute reward, the triplet formulation enables a **relative success criterion**: if

$$\text{score}(A, C) > \text{score}(A, B), \tag{3}$$

the polishing step is considered an improvement over the initial rendering.

This dual use of *absolute scoring* and *relative comparison* ensures that the model not only maximizes fidelity to the reference but also consistently outperforms its own initial generations, leading to more stable and effective RL optimization.

---

**Prompt**

You will be given three images:

- The first image is the reference design (target screenshot).

- The second and third images are code renderings generated based on the reference.

Your task is as follows:

1. Assign a similarity score (0–100) to both the second and third images with respect to the reference: - 0 = completely dissimilar.

- 100 = perfectly identical.

- When scoring, consider the following dimensions with approximate weights:

- Layout structure (30%): element positions, alignment, and overall layout.

- Color fidelity (25%): background, text, button colors, etc.

- Typography (20%): font size, weight, spacing, line height, etc.

- Spacing ratios (15%): margins, paddings, and spacing between elements.

- Element details (10%): button corners, borders, icon styles, etc.

- Ignore differences in actual image content (e.g., photos, icons), and only evaluate style fidelity.

2. Provide a brief justification for each score: - List 2–3 major differences and explain why they affect the score.

- If the rendering is highly consistent, state the reasons (e.g., "layout and colors are almost identical").

3. Provide a final conclusion: indicate which rendering (second or third) is closer to the reference. - The conclusion **must** be enclosed in LaTeX `\\boxed{}`.

- For example: `\\boxed{The second image is better}`

4. The output format must strictly follow this template:

```
Second image score: <score>
Reason: <brief explanation>

Third image score: <score>
Reason: <brief explanation>

\boxed{<which image is better>}
```

---

### A.3 SFT DATA CURATION DETAILS

To automatically and correctly build the SFT dataset, we carefully design task-specific data construction strategies. To ensure data fidelity which is a major bottleneck in UI generation, we employ a "reverse-engineering" strategy using state-of-the-art models (e.g., Claude-3.5-Sonnet, GLM-4.5V (Hong et al., 2025)).

For UI polishing, we diversify rendered inputs using multiple VLMs (our model, GLM-4.5V (Hong et al., 2025), Claude-4-Sonnet) and derive reasoning traces via VLM-generated comparisons rather than direct prompts, yielding more accurate rationales.

For UI editing, we cover addition, deletion, replacement, and adjustment operations, filter candidates with heuristic rules and manual checks, and address the difficulty of component addition by reversing high-quality deletion pairs. These details, though labor-intensive, ensure data diversity, precision, and reliability—reflecting the deliberate care invested in our SFT stage.

### A.4 Evaluation Metrics Specifications

#### A.4.1 Evaluation for UI-to-Code

For the UI-to-code task, we employ `o4-mini` as the visual evaluator to assess the fidelity of generated renderings. Given the reference screenshot $A$ and the rendering $B$ generated from the predicted HTML/CSS code, `o4-mini` outputs a similarity score $\mathrm{score}(A, B)$ in the range $[0, 100]$, where higher values indicate greater visual resemblance.

To obtain a robust evaluation metric, we define the final accuracy as the proportion of samples whose similarity score exceeds a threshold of 80:

$$\text{Accuracy} = \frac{1}{N} \sum_{i=1}^{N} \mathbb{1}\{\mathrm{score}(A_i, B_i) \geq 80\}, \tag{4}$$

where $N$ denotes the total number of evaluated UI examples. This threshold-based criterion ensures that only renderings with sufficiently high fidelity to the reference are considered successful.

The prompt provided to judge the similarity between the original UI screenshot and the rendering image is as follows:

---

**Prompt**

You will be given two images:
- The first image is the reference screenshot (design draft or target rendering).
- The second image is the rendering generated from the first image using HTML/CSS/frontend code.
Your task is to evaluate the similarity between the two images and assign a score on a scale from 0 to 100:
- 0 means completely dissimilar.
- 100 means perfectly identical.
The output must follow the required format:
1. Provide the final score, where the value **must** be enclosed in LaTeX \\boxed{}.
2. Provide a short justification, explaining the key similarities and differences that influenced your score.

---

#### A.4.2 Evaluation for UI Polishing

For the UI polishing task, we employ `Gemini-2.5-Pro` as the visual evaluator. The model is prompted with a triplet comparison: a reference screenshot $A$, an initial rendering $B$, and a polished rendering $C$. It is asked to assign similarity scores in the range $[0, 100]$ to both $B$ and $C$, provide brief reasoning for each score, and determine which rendering is closer to the reference.

The prompt template is shown below:



**Prompt**

You will be given three images:
- The first image is the reference (target design draft).
- The second and third images are code-rendered results based on the reference.
Please complete the following tasks:
1. Assign a score to both the second and third images, with a range of 0–100:
- 0 means completely dissimilar to the reference.
- 100 means exactly the same as the reference.
2. When scoring, consider layout, color scheme, typography, spacing, and element details.
3. Briefly explain the reason for each score.
4. Provide a final conclusion: which image is closer to the reference. The conclusion should be wrapped in LaTeX \\boxed{}, for example:

```
Second image score:  85
Reason:  Overall layout is consistent, but the font is
slightly smaller.  Colors are mostly accurate.
Third image score:  78
Reason:  Most elements are reproduced, but button styles and
spacing differ significantly.
\\boxed{The second image is better}
```



## A.5 DETAILS OF BENCHMARKS

Here we illustrate the details of benchmarks that we evaluate on, along with our curated *UIPolish-bench* and *UI2Code-Real*. To ensure a fair comparison between open-source and closed-source systems on our proposed benchmarks, we evaluate a diverse set of models. Specifically, we select 5 groups representative open-source VLMs, such as InternVL3 (Zhu et al., 2025), Qwen2.5-VL (Bai et al., 2025), MiMo-VL (Team et al., 2025a), Kimi-VL (Team et al., 2025b), and GLM-4.1V-9B-Thinking (Hong et al., 2025). For closed-source systems, we evaluate 4 widely-used models: Claude-4 (Anthropic, 2025), Gemini-2.5 (Comanici et al., 2025), Doubao (Guo et al., 2025), and GPT-5 OpenAI (2025). This setup allows us to benchmark UI-to-code and UI polishing performance across both research and industrial systems under the same evaluation protocol.

### A.5.1 EXISTING BENCHMARKS

- Web2Code (Yun et al., 2024): this benchmark comprises 1,198 webpage screenshot images to evaluate the ability of HTML code generation for a multi model. Different from traditional code-level evaluations, this benchmark assesses the generated webpage's fidelity at the image level. This evaluation method converts the predicted HTML codes back into images using Selenium WebDriver to allow a direct visual comparison with the ground truth images.

- Flame-React-Eval (Ge et al., 2025): a benchmark of 80 curated design-to-React cases. In the original evaluation, the generated code is judged correct if it compiles, renders without error, and the rendered screenshot matches the reference with a DINOv2 embedding cosine similarity above threshold.

- Design2Code (Si et al., 2024): contains 484 real-world webpages (plus an 80-example HARD subset) as input screenshots. Models must output corresponding HTML/CSS. The original evaluation is done via rendered visual similarity (CLIP) plus element-level matching (position, text, color), with human judgments used to validate metrics.

### A.5.2 OUR PROPOSED BENCHMARKS

Almost all the existing benchmarks are constructed with synthetic or heavily pruned HTMLs, and none of them can evaluate the UI-polish ability. To analyze the UI-to-code and UI-polish capability on real-world webpage distribution, we propose the following benchmarks.

- UI-to-code-Real: A benchmark consisting of 115 real-world webpage screenshots. Unlike synthetic datasets, which typically feature simplified layouts and over-pruned structures,

UI2Code-Real directly reflects the complexity, visual diversity, and noise inherent in real webpages. This benchmark therefore provides a more realistic and challenging setting for evaluating UI-to-code generation models.

- UIPolish-bench: A benchmark specifically designed to evaluate UI polishing. Each sample consists of a reference screenshot $A$, an initial rendering $B$, and the corresponding HTML/CSS code used to produce $B$. The goal of UI polishing is to compare $A$ and $B$, identify the discrepancies between them, and modify the underlying HTML/CSS code so that the rendered result better aligns with $A$. This design directly captures the iterative refinement process of UI development. UIPolish-Bench is further divided into two subsets: 1) **UIPolish-Synthetic**: constructed from synthetic webpages with controlled structures, which ensures clean annotations and facilitates fine-grained evaluation of polishing behavior. 2) **UIPolish-Real**: collected from real-world webpages, which preserves noise, complex layouts, and design diversity, providing a challenging benchmark for assessing polishing in practical settings.

### A.6  DEMO CASES

To provide an intuitive understanding of the proposed UI2Code$^N$, we present several representative demo cases focusing on UI-to-code and UI Editing:

- **UI-to-Code**: Given a raw UI screenshot, the model automatically generates executable HTML/CSS code that faithfully reproduces the layout, color scheme, and visual elements of the design. The demos show that our model is able to handle both simple layouts and complex, nested structures with high fidelity.

- **UI Editing**: Starting from an existing rendering, the model is able to perform targeted edits such as modifying layout alignment, adjusting typography, changing color themes, or inserting new components. These cases demonstrate the model's ability to act as an interactive assistant in iterative design workflows.

These demo cases highlight the versatility of our system across different aspects of UI development, demonstrating its potential as both a code generator and an interactive design assistant.

### A.6.1 CASES OF UI2CODE

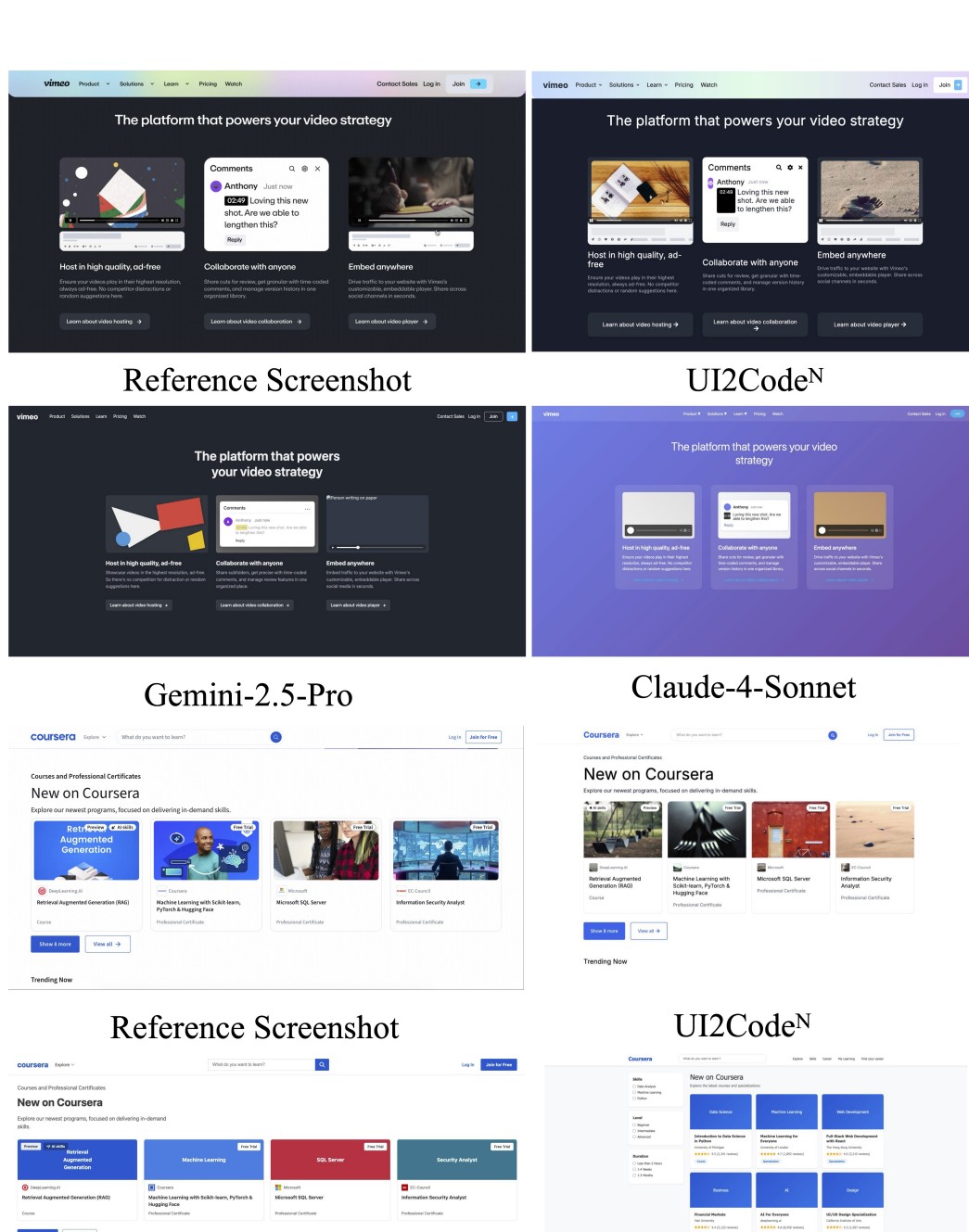

Figure 2: UI2Code$^N$ Demo Cases: UI-to-code (1/4)

Reference Screenshot  UI2CodeN

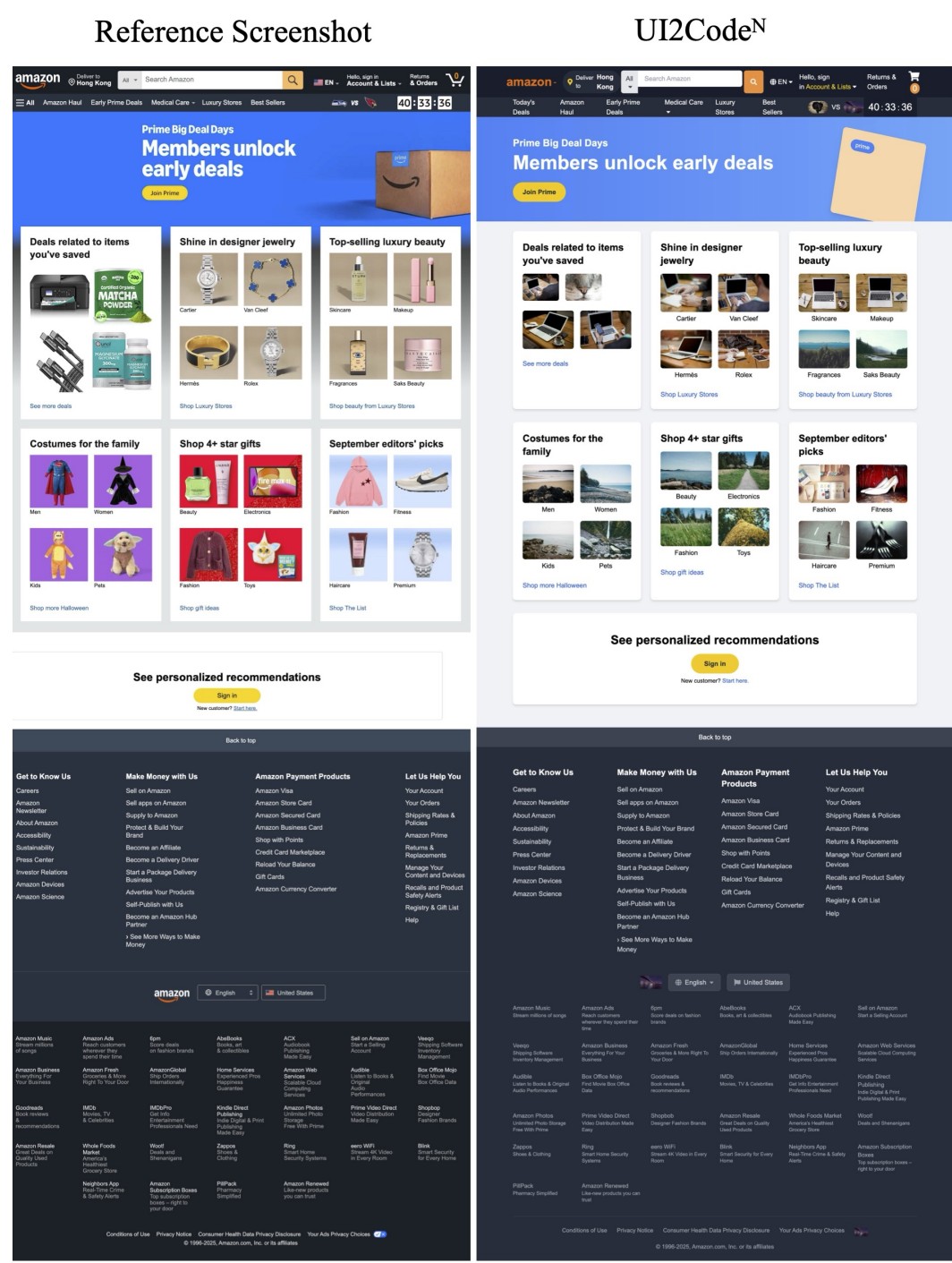

Figure 3: UI2CodeN Demo Cases: UI-to-code (2/4)

Reference Screenshot

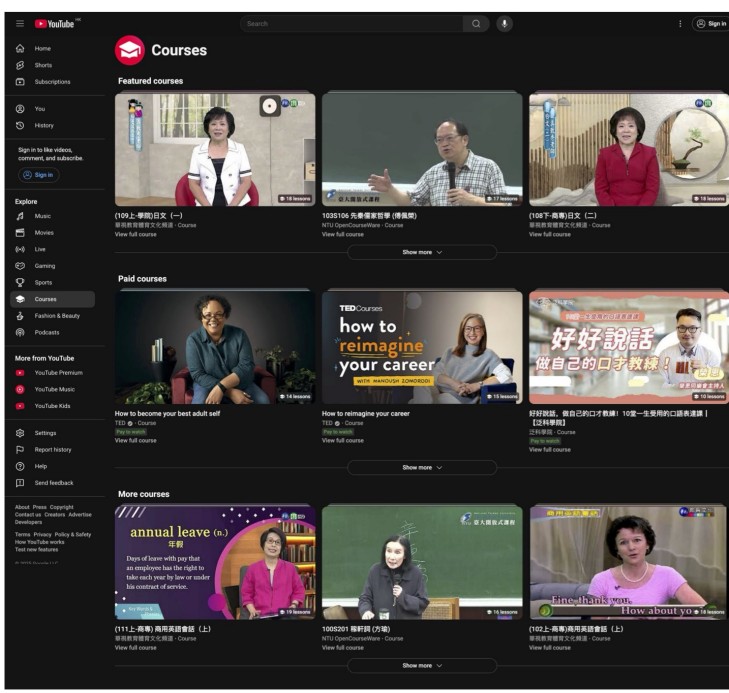

UI2Code$^N$

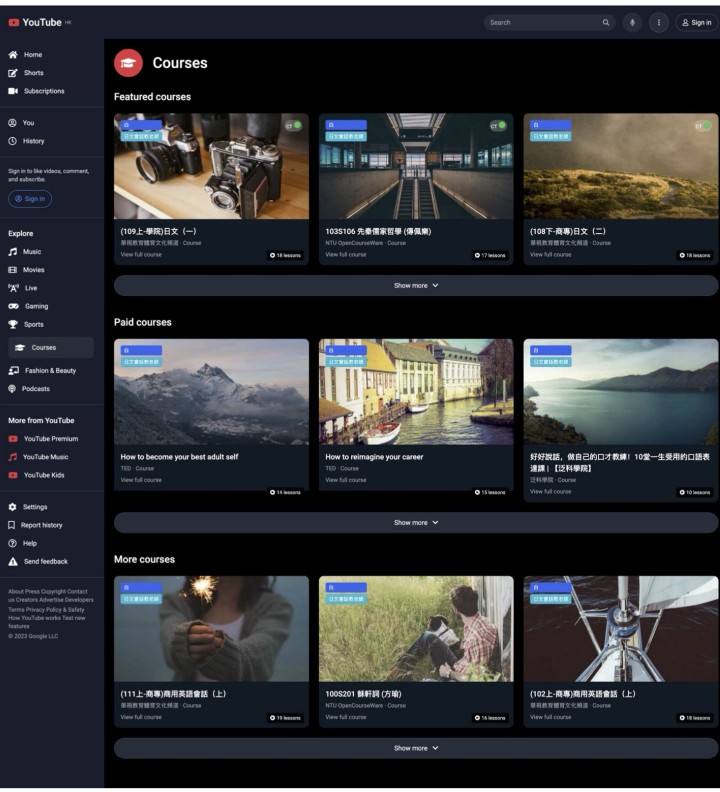

Figure 4: UI2Code$^N$ Demo Cases: UI-to-code (3/4)

Reference Screenshot                          UI2Code$^N$

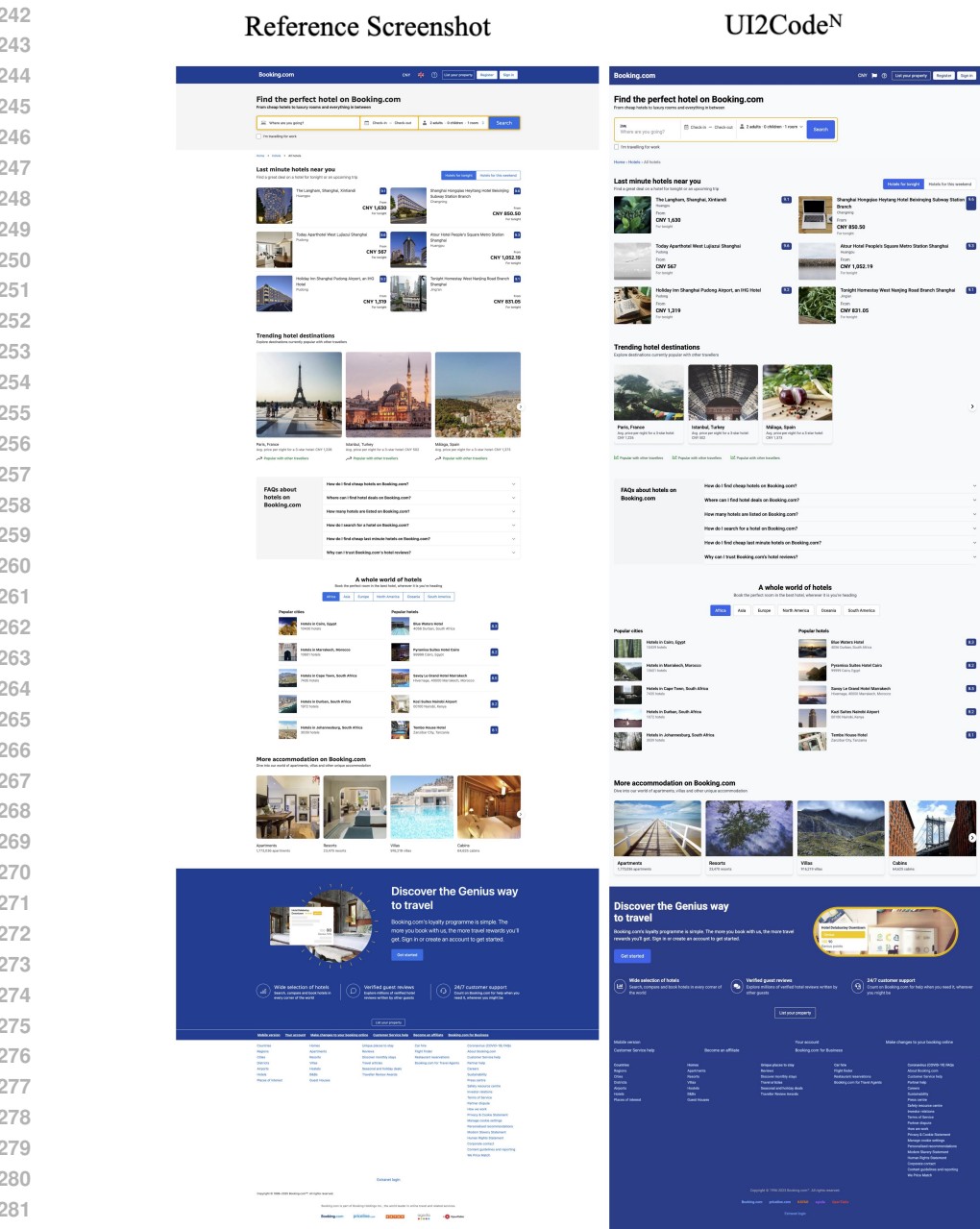

Figure 5: UI2Code$^N$ Demo Cases: UI-to-code (4/4)

### A.6.2 CASES OF UI EDITING

UI Editing: Change the news on the top to a Christmas story.

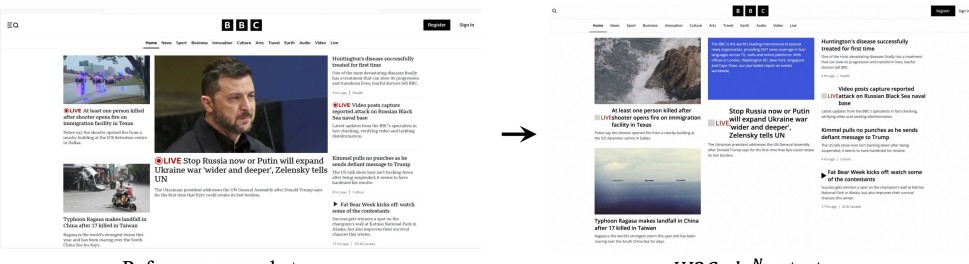

Reference screenshot.        $UI2Code^N$ output.

UI Editing: Delete the facebook login button.

Reference screenshot.        $UI2Code^N$ output.

UI Editing: Replace the photo of Zelensky with a short paragraph describing BBC on a blue background.

Reference screenshot.        $UI2Code^N$ output.

UI Editing: Align the 'New on Coursera' texts to the right side.

Reference screenshot.        $UI2Code^N$ output.

Figure 6: UI2Code$^N$ Demo Cases: UI Editing (1/2)

UI Editing: Change all blue elements into red.

UI Editing: Rearrange the layout of course cards to 2* 2.

UI Editing: Insert an author card with name of Yoshua Bengio.

UI Editing: Change the background color into yellow.

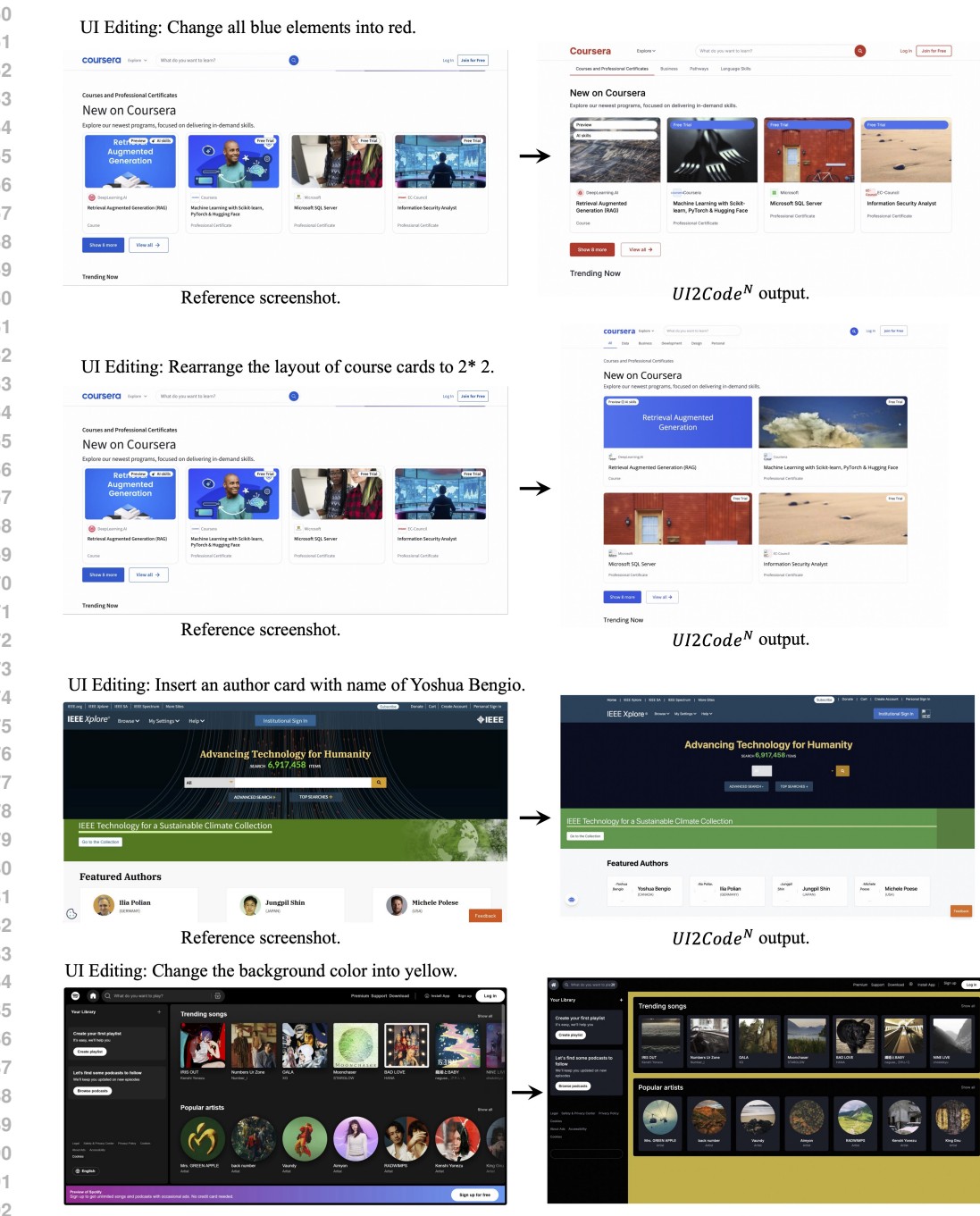

Figure 7: UI2Code$^N$ Demo Cases: UI Editing (2/2)

