# OpenReview forum: "UI2Code$^N$: A Visual Language Model for Test-Time Scalable Interactive UI-to-Code Generation"
_ICLR.cc/2026/Conference — Submitted to ICLR 2026_

### Official Review · Reviewer_8XSC · 2025-10-26

**Soundness:** 2
**Presentation:** 3
**Contribution:** 3
**Rating:** 2
**Confidence:** 4

**Summary:**

The paper proposes an interactive paradigm for UI-to-code which contains three capabilities: (1) UI-to-code draft generation, (2) UI polishing using target screenshot + draft code + rendered output, and (3) instruction-driven UI editing. The proposed model, UI2CodeN (9B), is trained via a three-stage pipeline: continual pretraining on large-scale crawled webpage pairs, supervised fine-tuning to seed the three tasks, and RL with a verifier and round-robin comparator reward. The authors argue this enables test-time scaling: iteratively polishing improves fidelity (they report ~12% gain with four rounds).

Empirically, UI2CodeN achieves strong results across Design2Code, Flame-React-Eval, Web2Code, and two new benchmarks (UI2Code-Real, UIPolish-bench), according to VLM evaluators.

**Strengths:**

- The paper introduces an end-to-end training recipe containing three distinct stages, continual pretraining + SFT + RL, to align VLMs with multimodal UI code generation.
- UI2CodeN achieves new state of the art among open VLMs on multiple benchmarks; competitive vs. top closed models according to VLM evaluators. Evaluation results exhibit test-time scaling capabilities.
- Contributes two new benchmarks for UI2Code generation: UIPolish-bench and UI2Code-Real.

**Weaknesses:**

- Heavy reliance on VLM-based judges for reward signals and evaluation, which can introduce biases and reward gaming risks. While the paper has shown efforts in calibrating rewards with comparator + round-robin, there's no evaluation on whether the GLM-4.5V based reward mechanism aligns with human judgments, or whether using different VLM evaluators provides consistent judgments.

- All benchmarks are evaluated solely via VLM evaluators (o4-mini for UI-to-code and Gemini-2.5-Pro for UI polishing), without reporting any human evaluation or the original, non-VLM metrics for the cited benchmarks, making the validity and significance of the reported performance highly questionable.

- In reward design, GLM-4.5V is finetuned with SFT to improve robustness, but the paper does not disclose the training procedure, data curation, or any metric or evidence to support the claimed robustness.

- Missing citation for relevant literature “Sketch2code: Evaluating vision-language models for interactive web design prototyping”, which proposed a similar iterative UI-to-code generation framework.

- Data governance: ~10M crawled UI-code pairs seeded from Common Crawl; the paper does not detail licensing, PII filtering, or copyrighted asset handling.

- The comparator + round-robin scheme incurs O(N^2) VLM calls per rollout, casting doubt on the scalability of the proposed RL approach.

- The claim UI Editing capabilities are shown only through demos, without formal evaluations.


While the paper proposed an ambitious approach to train interactive UI-to-Code reasoning models at scale with test-time scaling, the authors need to revise the soundness and transparency of their method and evaluation protocol in order for this work to be accepted by the ICLR venue.

**Questions:**

- The paper reported using o4-mini for UI-to-code benchmarks but gemini-2.5-pro for UI polishing benchmarks. Any justification for why these two specific VLMs are chosen for the two evaluation tasks?
- How sensitive are results to evaluator choice and prompt wording? Do different VLM judges produce consistent/calibrated outputs? Please report results with at least one independent open judge and a human study (N≥50) on Design2Code-HARD.
- Did you test RL-trained models using a held-out verifier never seen during RL (e.g., DINOv3-based metric, IoU-based block/element matching, human preferences)? Any failure cases where visual similarity rises but DOM/typography regress?
- Can the authors perform ablation studies on the continual pretraining? Given a strong base model, would SFT + RL alone give competitive performance even without the pretraining stage?
- Do the authors observe any oscillations or quality regressions across the rounds?
- Can the authors report any quantitative or qualitative evaluations for UI editing capabilities of UI2CodeN against strong baselines?
- Please detail crawl sources, robots/ToS compliance, PII filtering, and license auditing; do the authors plan to release any of the curated datasets?

---

> ### Author Response · Authors · 2025-11-26
> **Rebuttal (1/5)**
>
> Dear Reviewer,
>
> Thank you for the thoughtful and constructive feedback. We appreciate the reviewer’s recognition of the ambition and potential impact of our approach to scaling interactive UI-to-Code reasoning. We fully acknowledge the importance of strengthening the soundness and transparency of our evaluation protocol, and have carefully addressed these concerns.
>
> ---
>
> ### **Response to W1: Alignment of VLM-based Rewards with Human Judgments and Sensitivity across VLMs**
>
> See the **general response** for "2. Alignment analysis between human evaluators and VLM evaluators". Additional analysis on **evaluator sensitivity** is provided in **Response to Q2.**
>
> ---
>
> ### **Response to W2: Validity of Evaluation Metrics (Human Eval & Traditional Metrics)**
>
> See the **general response:** "1. Human evaluation results", "4. Cross-verification with additional held-out evaluators".
>
> ---
>
> ### **Response to W3:  Details of Reward Model Training & Robustness Verification**
>
> In the revised Appendix A.2, we have now provided a comprehensive disclosure of the Model Selection, Data Curation, Training Dynamics,, and Robustness Verification.
>
> 1. **Rationale for Model Selection & SFT**
>
>    We selected GLM-4.5V as the base model due to its open-source nature and superior local deployment capabilities (ensuring reproducibility, stability and query bandwidth). Our preliminary study revealed a specific capability gap: while the base GLM-4.5V performed satisfactorily at *absolute scoring* (whether the score is >=80 or <80), it struggled with *relative judgment* on triplets(ranking two UIs with similar performance compared to the reference UI).
>
>    This motivated our decision to apply Supervised Fine-Tuning (SFT) specifically to bridge this gap.
>
> 2. **Rigorous Data Curation**
>
>    To ensure high-quality reward signals, we adopted a knowledge distillation strategy:
>    - Source: We sampled 10k UI-Code triplets (reference UI, generated-UI-1, generated-UI-2), with generated UI sampled from diverse UI-to-Code models.
>    - Teacher Labeling: We utilized Gemini-2.5-Pro as a teacher model to label these pairs based on visual alignment and code correctness.
>    - Quality Control: We filtered out ambiguous samples (low teacher confidence), yielding a balanced, high-quality dataset of 8,500 pairs for relative preference learning.
>
> 3. **Training Procedure**
>
>    We fine-tuned the model to classify the better UIs using standard Cross-Entropy Loss, and SFT with a learning rate of 2e-5 and a packed batch size of 64 with sequence length of 32K for 200 iterations.
>
> 4. **Evidence of Robustness (New Ablation Study)**
>
>    To quantitatively demonstrate the benefit of SFT and verify robustness, we evaluated the **Human-Model Agreement of relative judgement on triplets** on a held-out validation set (100 samples). We included a state-of-the-art commercial model (GPT-5) as a high-level reference. The significant leap from 62% to 94% demonstrates that our SFT process effectively ensures a robust and accurate reward signal, outperforming even much larger generalist models on this specific task.
>
> | Model                     | Agreement with human |                  Interpretation                  |
> | :------------------------ | :---------------------------------------------------: | :----------------------------------------------: |
> | Base GLM-4.5V (Zero-shot) |                          62%                          | Shows the "domain gap" and the necessity of SFT. |
> | Commercial SOTA (GPT-5)   |                          85%                          |    Strong generalist performance (Reference).    |
> | Ours (SFT GLM-4.5V)       |                          94%                          |  Surpasses SOTA via domain-specific alignment.   |
>
>
>
> ### **Response to W4: Inclusion of Relevant Literature "Sketch2code"**
>
> We thank the reviewer for pointing out this relevant literature. We have cited the suggested work (Sketch2code) in the revised Related Work section to acknowledge its context within the field of VLM-based UI coding.

---

> ### Author Response · Authors · 2025-11-26
> **Rebuttal (2/5)**
>
> ### **Response to W5 & Q7: Ethics / Data Governance**
>
> We appreciate the reviewer for raising these important ethical and legal concerns. We clarify the data governance pipeline below, and have included a dedicated **Ethics Statement section** in the revision to ensure full transparency.
>
> - **Crawl Sources.** Common Crawl is used only as a URL index to identify publicly accessible webpages. All HTML/CSS content and UI screenshots used for training are obtained solely through our own crawler; we do not use or redistribute Common Crawl’s archived content.
> - **Robots/ToS Compliance.** Our crawler strictly respects site-level policies:
>   - `robots.txt` is enforced for every domain,
>   - domains whose Terms of Service prohibit crawling, reuse, or derivative processing are excluded,
>   - login-gated, paywalled, or personalized pages are filtered out, and
>   - conservative rate limits prevent unintended server load.
> - **PII Filtering.** Before training, all collected webpages undergo automatic PII filtering (removal of emails, phone numbers, user tokens, names, or other sensitive user traces) and UI-level sanitization to exclude pages containing personal or sensitive information.  Pages failing these checks are discarded prior to training.
> - **License Auditing and Copyright Mitigation.**  We explicitly acknowledge that webpage content is subject to third-party rights. To mitigate risk:
>   - domains with restrictive licenses or prohibitive ToS are excluded,
>   - Proprietary images, icons, and branded visual assets are filtered,
>   - and no raw webpage content (screenshots or source code) is ever released. The model learns structural UI-to-code patterns rather than reproducing page-level copyrighted material.
> - **Dataset Release.** To avoid privacy and copyright concerns, we do not release any part of the curated dataset.
>   -  Only the trained model (built upon an Apache-2.0 licensed base model) and evaluation code will be released for **research-only, non-commercial use**.
>
> These clarifications will be included in the revised manuscript. We believe this fully addresses the reviewer’s concerns regarding data governance, privacy, and legal compliance.
>
>
> ---
>
>
> ### **Response to W6: Scalability of the Comparator Reward Design**
>
> While the round-robin comparator is formally O(N²) within a rollout group, we argue that this **does not impact the actual scalability** of our RL method for three concrete reasons:
>
> **TL;DR:** *Although the comparator is formally O(N²), N is small, fixed, and not scaled; moreover, the effective computation is further reduced by validity filtering and prefix-level optimizations. Thus, the round-robin reward remains practically efficient and does not impede large-scale RL training.*
>
> 1. **N is a fixed small hyperparameter, not a scaling dimension.** Scalability in LLM RL is dominated by *number of prompts*, *training steps*, and *model size*. The rollout count per prompt (N) is not scaled up in practice. Standard and effective RL settings (e.g., GRPO, DAPO) use N = 4 ~ 16; we use N = 8. Thus the "O(N²)" term is, in effect, a small constant cost, not a barrier to scaling.
> 2. **The absolute overhead is negligible compared to generation.** Even with a full round-robin at N = 8, only 28 VLM comparisons are needed. Autoregressive UI-code generation is substantially more expensive (long contexts, multi-step decoding). Empirically, these comparisons do not appear as a bottleneck in our training loop.
> 3. **The effective comparison set is much smaller in practice.** Before invoking the VLM, we automatically reject samples that fail to render or produce invalid outputs. Only valid samples (M ≤ N) enter the comparator, reducing the cost to O(M²). Early in training or on harder tasks, M is often significantly smaller than N.
> 4. **Engineering optimizations further shrink the cost.** All comparisons share the same long prefix, making reward inference highly cache-friendly. Prefix caching or batched VLM evaluation substantially reduces real compute even within the small N regime.
>
> ---
>
> ### **Response to W7: UI Editing Evaluation**
>
> See the **general response:** "1.2 Human evaluation on UI-Editing".

---

> ### Author Response · Authors · 2025-11-26
> **Rebuttal (3/5)**
>
> ### **Response to Q1: Why Different VLMs for Evaluation**
>
> **Overall Conclusion**: We selected different models for the two tasks not arbitrarily, but based on the distinct capability requirements of each task and our empirical findings. Our experiments demonstrate that GPT-o4-mini is a cost-effective yet fully sufficient evaluator for pairwise similarity, whereas Gemini-2.5-Pro is uniquely capable of achieving high alignment with human judgments in the more demanding triplet comparison setting.
>
> The detailed justification is as follows:
>
> 1. **Task-Specific Capability Requirements:**
>
>    1. **UI-to-Code Generation:** This task involves a direct pairwise comparison between *the Reference Screenshot and the Rendered Image*.
>
>    2. **UI Polishing:** This is a more complex triplet comparison involving three images: *the Reference Screenshot, the First Render, and the Second (Polished) Render*. The goal is to determine if the Second Render represents a relative improvement over the First towards the Reference.
>
> 2. **Empirical Selection via Human Alignment:**
>
>    1. GPT-o4-mini for **UI-to-Code Generation:** To validate the reliability of GPT-4o-mini for this task, we conducted a human evaluation on 100 randomly sampled cases from the Design2Code dataset. GPT-o4-mini demonstrated a 92% agreement rate with human evaluators. Moreover, o4-mini and Gemini exhibit a very high correlation in their scores on this task (r = 0.9998), indicating that using a more expensive model would not materially change the conclusions while significantly increasing cost. We therefore adopt GPT-o4-mini as the most cost-effective choice for UI-to-Code.
>
>    2. Gemini-2.5-Pro for **UI Polishing:** We initially attempted to use GPT-o4-mini for the Polishing benchmark. However, our qualitative analysis revealed that these models struggled to accurately judge subtle visual improvements in the triplet setting (often failing to distinguish which render was "closer" to the reference). We subsequently conducted a pilot study comparing various VLMs against human rankings. Results of human agreement on UI polistion:
>       - GPT-o4-mini: 77%
>       - GPT-5: 85%
>       - Gemini-2.5-Pro (Final selection): 94%
>
>    The results show that Gemini-2.5-Pro achieved a significantly higher correlation with human judgment (94% agreement) compared to GPT-4o-mini (77% agreement) for detecting these nuanced visual improvements.
>
> ---
>
> ### **Response to Q2: Sensitivity of Evaluator**
>
> We appreciate the reviewer's rigorous scrutiny regarding the robustness of our evaluation metrics. We have conducted additional experiments to address concerns about evaluator sensitivity and consistency.
>
> Across four diverse evaluators (o4-mini, GLM-4.5V, Claude-4-Sonnet, Gemini-2.5-Pro), we observe perfect **rank-order consistency** (GPT-5 > Ours-RL > Ours-SFT > Qwen2.5-VL-72B) and **highly stable score trends (Pearson r > 0.98)**. Moreover, the **relative performance gaps are also consistent**: GPT-5 and Ours-RL are always the closest, while Qwen2.5-VL-72B exhibits the largest gap to all other models. The RL model **consistently improves** over the SFT baseline by + 6~10 points across all evaluators, demonstrating that the gains are evaluator-invariant rather than tied to any specific judge.
>
> |     Model      | o4-mini |      | GLM-4.5V |      | Claude-4-Sonnet |      | Gemini-2.5-pro |      |
> | :------------: | :-----: | :--: | :------: | :--: | :-------------: | :--: | :------------: | :--: |
> |                | Score ↑ | Rank | Score ↑  | Rank |     Score ↑     | Rank |    Score ↑     | Rank |
> | Qwen2.5-VL-72B |  41.9   |  4   |   56.6   |  4   |      33.5       |  4   |      33.1      |  4   |
> |     GPT-5      |  89.7   |  1   |   93.2   |  1   |      84.7       |  1   |      82.6      |  1   |
> |    ours-SFT    |  79.3   |  3   |   84.7   |  3   |      75.4       |  3   |      72.7      |  3   |
> |    ours-RL     |  88.6   |  2   |   91.1   |  2   |      84.5       |  2   |      82.2      |  2   |

---

> ### Author Response · Authors · 2025-11-26
> **Rebuttal (4/5)**
>
> ### **Response to Q3: Held-out Evaluation Verifiers Between SFT and RL**
>
> **Part1**: "*test RL-trained models using a held-out verifier never seen during RL*".
>
> See **General Response** "4. Cross-Verification with Additional Held-out Evaluators".
>
> **Part2**: "*Any failure cases where visual similarity rises but DOM/typography regress?*"
>
> To address the concern regarding visual improvements might come at the cost of structural degradation, we evaluated the models using a diverse set of held-out verifiers. These metrics go beyond visual similarity, assessing the generated code's structural integrity, content accuracy, and layout precision.
>
> The results indicate that the significant improvement in VLM reward during RL training does not compromise the underlying code structure. Instead, When the VLM-based visual score surged from 79.3 to 88.6, **the structural metrics (Block and Position) also saw consistent gains (Block: +1.9, Position: +2.1)**, validating the robustness of our RL approach.
>
>
>
> | **Metric** level | **Metric**        | **our-SFT** | **our-RL** | **Gain** |
> | ---------------- | ----------------- | ----------- | ---------- | -------- |
> | Code-Level       | Block Accuracy    | 86.8        | 88.7       | **+1.9** |
> | Code-Level       | Text Accuracy     | 91.5        | 93.1       | **+1.6** |
> | Code-Level       | Position Accuracy | 81.7        | 83.8       | **+2.1** |
> | Code-Level       | Color Consistency | 69.7        | 72.6       | **+2.9** |
> | Visual-Level     | VLM-based Score   | 79.3        | 88.6       | **+9.3** |
>
> ---
>
> ### **Response to Q4: Necessity of Continual Pretraining**
>
> We conducted the suggested ablation study, and the results provide strong evidence that the pretraining stage plays a critical role. As shown in the table below, **removing the pretraining stage results in an absolute accuracy drop of 8.5%**, clearly indicating that even with a strong base model, SFT alone is insufficient to bridge the domain gap.
>
> Most state-of-the-art open-source VLMs, such as Qwen2.5-VL [1] and InternVL3 [2], construct their pretraining data primarily from image captioning, OCR, grounding, video, and general text-only datasets. However, these mixtures crucially lack aligned image–code pairs. As a consequence, the base models do not acquire the fine-grained cross-modal alignment required to directly map visual layouts to structured, nested DOM representations.
>
> Our Continual Pretraining (CPT) stage is explicitly designed to address this limitation by injecting domain-specific syntax and layout knowledge, thereby enabling effective visual-to-DOM alignment. The ablation results strongly validate the necessity of this targeted pretraining stage.
>
> |        Setting         | Design2Code Accuracy (%) |
> | :--------------------: | :----------------------: |
> | SFT w/ pretrain (Ours) |           72.1           |
> |    SFT w/o pretrain    |           63.6           |
>
> [1] Bai, Shuai, et al. "Qwen2. 5-vl technical report." *arXiv preprint arXiv:2502.13923* (2025).
>
> [2] Zhu, Jinguo, et al. "Internvl3: Exploring advanced training and test-time recipes for open-source multimodal models." *arXiv preprint arXiv:2504.10479* (2025).

---

> ### Author Response · Authors · 2025-11-26
> **Rebuttal (5/5)**
>
> ### **Response to Q5: Oscillations of Polishing Process**
>
> We appreciate this insightful question. **From a perspective of macroscopic stability**, the overall performance metrics show a steady upward trend across rounds, as demonstrated in Table 2 in our manuscript. This indicates that the refinement process is globally effective. **From a microscopic perspective**, we observed oscillations and quality regressions in specific instances, although the overall trend remains positive.
>
> We have conducted a fine-grained statistical analysis to investigate this phenomenon:
>
> 1. **Polish Accuracy Across Rounds**
>
> | Polish Round | Polish Accuracy |
> | :----------: | :-------------: |
> |      1       |      66.0       |
> |      2       |      64.7       |
> |      3       |      63.3       |
> |      4       |      65.8       |
>
> 2. **Individual-level Oscillations**
>
> To investigate the root cause of these fluctuations, we stratified samples based on their initial UI2Code scores using 80 as a convenient split point. This threshold does not imply any intrinsic task difficulty; it simply separates higher-scoring cases (where the initial output is already close to acceptable (simple cases)) from lower-scoring cases (where more substantial issues remain (complex cases)). We then calculated the "Polish Success Rate" (probability of score improvement in the next round).
>
>   **High Effectiveness on Complex Cases (< 80):** For cases with lower initial quality, the model demonstrates a remarkably high success rate of 74.5%. This confirms that our refinement method is highly effective at identifying and fixing substantive errors (e.g., layout structure, missing components).
>
>   **Convergence on High-Quality Cases (>= 80):** For cases that are already high-quality, the success rate naturally drops to 53.1%. This indicates that the generated code has reached the model's capability ceiling. At this stage, further "polishing" yields diminishing returns, where the model may explore stylistic variations (oscillations) rather than functional improvements.
>
>   | UI2Code Score Range | Polish Success Rate |
> | :-----------------: | :-----------------: |
> |        >= 80         |        53.1         |
> |        < 80         |        74.5         |
>
>
>
> ---
>
>
> ### **Response to Q6: Evaluation of UI Editing Capabilities**
>
> See **general response**: "1.2 Human evaluation on UI-Editing".

---

> ### Author Response · Authors · 2025-11-27
> **Rebuttal — Newly Added Supplementary Material: Qualitative Comparison on Design2Code-HARD**
>
> Dear Reviewer,
>
> Thank you again for your feedback. Following your suggestions, we have made substantial improvements to the evaluation protocol, ablation studies, method presentation and analysis, as well as the ethics statement throughout the revised rebuttal.
>
> **As an additional contribution in this rebuttal, we newly include supplementary materials** that provide qualitative generation examples of **UI2Code^N-rl** on **Design2Code-HARD**, aiming to better reflect real-world UI-to-code generation performance. These examples are presented alongside results from both closed-source and open-source state-of-the-art models, including **GPT-5, Gemini-2.5-Pro, Claude-Sonnet-4, and Qwen2.5-VL-72B**.
>
> In addition, we conduct **human evaluation** on Design2Code-HARD to compare overall generation quality. A **visualization** of human evaluation results is available at **https://postimg.cc/30d21V8q**, while the **complete results and a more detailed analysis** are provided in the **General Response** section.
>
> | Model Comparison         | **Win** | **Tie** | **Lose** | **Win + Tie** |
> | ------------------------ | :-----: | :-----: | :------: | :-----------: |
> | Ours vs. GPT-5           |  56.3%  |  16.3%  |  27.5%   |     72.5%     |
> | Ours vs. Gemini-2.5-Pro  |  52.5%  |  12.5%  |  35.0%   |     65.0%     |
> | Ours vs. Claude-Sonnet-4 |  63.8%  |  16.3%  |  20.0%   |     80.0%     |
> | Ours vs. Qwen2.5-VL-72B  |  96.3%  |  2.5%   |   1.3%   |     98.8%     |
>
> Overall, we believe that the newly added supplementary qualitative results and the corresponding human evaluation comprehensively address the reviewer’s concerns regarding evaluation completeness and real-world applicability, and further demonstrate the robustness and effectiveness of UI2Code^N.
>
> **We sincerely appreciate your time and thoughtful feedback, and we are happy to provide further clarification if needed.**

---

### Official Review · Reviewer_q7LE · 2025-10-27

**Soundness:** 3
**Presentation:** 3
**Contribution:** 3
**Rating:** 6
**Confidence:** 4

**Summary:**

The authors introduce UI2Code$^N$, a 9B-parameter open-source visual language model (VLM) tailored for interactive UI-to-code generation. Unlike traditional one-shot models, UI2Code$^N$ operates under a novel Interactive UI-to-Code paradigm encompassing three tasks: UI-to-code generation, UI polishing, and UI editing. The model is trained through a multi-stage process: large-scale pretraining on noisy web data, supervised fine-tuning on curated HTML datasets, and reinforcement learning (RL) with VLM-based reward functions. Experiments across benchmarks (Design2Code, Flame, Web2Code) and new curated datasets (UI2Code-Real, UIPolish-bench) demonstrate state-of-the-art performance for open-source models, with performance rivaling or surpassing leading closed-source models such as GPT-5 and Gemini-2.5-Pro. Notably, the model achieves up to 94% accuracy on synthetic UI polishing and demonstrates significant gains with test-time scaling (up to 12% improvement after 5 interaction rounds).

**Strengths:**

1. **Novel paradigm for interactive UI coding.** The proposed interactive paradigm realistically models the iterative design process of developers, introducing three interlinked capabilities (UI-to-code, polishing, editing) that improve practicality and performance.
2. **Comprehensive multi-stage training pipeline.** The model benefits from a carefully designed training strategy combining real-world pretraining, high-quality fine-tuning, and reinforcement learning with a sophisticated VLM-based reward design.
3. **Strong empirical results with new benchmarks.** The model outperforms prior open-source models across all tasks and matches or outperforms closed-source ones.

**Weaknesses:**

1. **Missing comparison with recent agent-based or modular systems.** While agent-style approaches (e.g., DECLARUI, DCGen, ScreenCoder) are mentioned in Sec 2. related work, direct comparisons with these systems are absent, which could provide insights into trade-offs between agent complexity and VLM-based simplicity.
2. **Lack of human evaluation.** All evaluations are based on VLM scoring or CLIP metrics. Human evaluations could validate whether the improvements translate into actual perceived quality and usability, especially in edge cases where automated scores may be misleading.
3. **Lack of UI editing evaluation.** Although UI editing is listed as one of the three core capabilities (UI-to-code generation / UI editing/ UI polishing). The paper lacks a detailed evaluation section or results table explicitly analyzing editing compared to other models, except for Appendix A.5 qualitative examples.

**Questions:**

see weaknesses

---

> ### Author Response · Authors · 2025-11-26
> **Rebuttal**
>
> We thank the reviewer for the positive assessment of our novel interactive paradigm and comprehensive training pipeline. We have carefully addressed the three concerns regarding baselines, human evaluation, and editing tasks with new experiments and analysis.
>
> ### **Response to Weakness 1: Comparison with Agent-based Systems**
>
> To compare against agent-bastd systems, we conducted additional experiments on the **Design2Code-HARD** benchmark (a challenging subset requiring complex layout understanding) , and choose recommended ScreenCoder and DCGen as the baseline.
>
> **Results:** our UI2Code^N-RL model outperforms DCGen significantly and surpasses the strong baseline ScreenCoder, particularly on the VLM-judge metric (+8.6%), which aligns better with human visual perception than pure structural metrics like DINO.
>
> **Performance Analysis:** We found that **agent-based systems suffer from error propagation**. For example, if the initial detection/OCR module fails to identify a small icon, the subsequent code generation step has no way to recover. In contrast, our VLM-based approach processes the UI holistically, making it more robust to complex layouts.
>
> **Efficiency Analysis**: The agent approach requires iterative interaction or multi-stage prompting, leading to **2-3$\times$ higher latency** and **2-3$\times$ token costs** compared to ours.
>
> | Model                                       | DINO-based Accuracy | VLM-judge Accuracy | Avg. Inference Latency (s) | Avg. Token Cost |
> | ------------------------------------------- | ------------------- | ------------------ | -------------------------- | --------------- |
> | DCGen (Agent-based systems)                 | 69.6                | 45.0               | ~137s                      | ~7600           |
> | ScreenCoder (Agent-based systems)           | 85.7                | 80.0               | ~66s                       | ~4600           |
> | UI2Code^N-RL (Ours,End-to-End) | 86.1                | 88.6               | ~40s                       | ~2600           |
>
> ### **Response to Weakness2: Human Evaluation**
>
> See the **general response**: "1. Human evaluation".
>
>
>
> ### **Response to Weakness3: UI Editing Evaluation**
>
> See the **general response**: "1.2 Human evaluation on UI-Editing".

---

### Official Review · Reviewer_3Wur · 2025-10-29

**Soundness:** 2
**Presentation:** 2
**Contribution:** 2
**Rating:** 2
**Confidence:** 4

**Summary:**

The paper introduces UI2CodeN for an interactive UI-to-code paradigm, which includes test-time scaling through multi-round polishing. Experiments show strong results over several VLMs. However, the main techniques are modest, such as standard prompt design and standard staged training, and the evaluation relies heavily on VLM-based scoring without human judgments or variance-aware reporting for this creative task.

**Strengths:**

1.	Clear problem framing that recasts UI-to-code as an interactive, multi-turn process with UI-to-code, polishing, and editing, which matches real workflows.
2.	Comprehensive training recipe with pretraining on web data, supervised fine-tuning with think/answer formatting, and RL with a comparator-style verifier.

**Weaknesses:**

1.	Missing crucial methodological details, including how the reward is computed and how the training loss function is designed. Detailed staged-training strategies are missing.
2.	The main techniques rely on standard pre-training, SFT, and RL process without designed training framework, a purpose-built loss or principled reward shaping objective for UI code fidelity, showing limited contributions.
3.	The evaluator’s reliability is not measured. Evaluation is heavily reliant on VLM-based scoring, and there is no human study for fidelity and any variance-aware reporting. The conclusions are not convincing, as this is a creative and subjective task.
4.	Limited comparisons of UI2Code relevant baselines. Experiments focus on vanilla VLMs without UI2Code advanced baselines, such as Uicoder [1] and ScreenAI [2], making it hard to attribute gains to the proposed paradigm.
5.	Limited ablation analysis. For the impact of each stage (pretraining vs SFT vs RL), the think/answer format, and the round-robin comparator, which components actually drive the improvements, and by how much?

[1] Uicoder: Finetuning large language models to generate user interface code through automated feedback. NAACL 2024.
[2] ScreenAI: A visual language model for UI and visually-situated. IJCAI 2024.

**Questions:**

1.	See above.
2.	It would be clearer to present the experimental settings in the Experiments section rather than in the Methods section.

---

> ### Author Response · Authors · 2025-11-26
> **Rebuttal (1/2)**
>
> Dear Reviewe,
>
> Thank you for your valuable feedback. We sincerely appreciate your insights and are fully committed to addressing these points to improve the quality of our work.
>
>
> ---
>
>
> ### **Response to W1: Clarification on Methodological Completeness & Enhanced Presentation**
>
> We appreciate the reviewer’s helpful suggestion regarding methodological clarity.
>
> 1. **Clearer Presentation in the Main Text**
> To further enhance readability, we now explicitly present all loss functions in Sec. 3.2.1, 3.2.2, and 3.2.3. A concise pseudo-code summary of the reward computation has also been added in Algorithm 1–3 (Sec. 3.2.3) to provide immediate intuition.
> 2. **Clarification of Existing Completeness**
>    Most of the methodological details mentioned by the reviewer were **already included in the original submission:**
>    1. Staged training strategies in Sec. 3.2.1 (Continual Pre-training), Sec. 3.2.2 (SFT), and Sec. 3.2.3 (RL).
>    2. Full pseudo-code and implementation details in Appendix A.1 and A.2, placed there due to space constraints.
>
> We hope these revisions make the methodology easier to follow while preserving the technical completeness of the original submission.
>
> ---
>
> ### **Response to W2: Novelty on Framework, Training Adaptations, and Reward Shaping**
>
> We appreciate the reviewer’s comment and would like to **clarify a significant misunderstanding:** The core contributions of our work are *not* the use of standard pre-training, SFT, or RL pipelines. These are merely the optimization backbone. Our technical novelty lies in (1) a purpose-built Iterative UI-to-Code Generation Framework, and (2) domain-specific objectives, data strategies, and (3) reward designs crafted explicitly for UI fidelity, none of which are standard or previously established.
>
> **1. An interactive and Iterative UI-to-Code Generation Framework**
>
> The statement that our method lacks a “designed training framework” overlooks our primary contribution. Our work introduces a new agentic formulation for UI coding, built around:
>
> - **Recursive, Visual-Feedback-Driven Workflow (Sec. 3.1):** Unlike traditional one-shot generation, our framework performs UI-to-code, UI-polish, and UI-edit in an iterative, self-improving loop driven by rendered visual feedback. This design mirrors real UI development workflows and enables test-time scaling, an ability impossible under standard SFT or RL alone.
> - **Domain-Specific Training Procedure (Sec. 3.2):** To support this framework, we introduce several UI-specific design choices:
>   - UI-grounded task and loss formulation, including a local alignment loss for fine-grained visual grounding.
>   - A tailored data pipeline, including large-scale UI corpus construction and automatic reverse-engineering synthesis during SFT.
>   - An RL setup designed specifically for polishing/editing behaviors, which arise only because of our iterative framework.
>
> These are not generic components: they are customized for the unique structure of UI coding and are necessary to make the iterative agentic pipeline viable.
>
> **2. Principled Reward Shaping Designed for UI Fidelity**
>
> The reviewer’s remark that we lack a “principled reward shaping objective” is inconsistent with the methods presented in the paper.
>
> - Identified Challenge: Standard metrics (pixel similarity, CLIP, single-VLM scoring) are unstable and misaligned for UI evaluation.
> - **Our Solution — Round-Robin Reward (Sec. 3.2.3):** We design a domain-specific reward mechanism that evolves through three steps, culminating in a Comparator +  Round-Robin reward design that aggregates preference signals across multiple candidates. This yields a robust, calibration-resistant reward that effectively guides GRPO.
> - **Ablation Evidence:** We provide controlled experiments showing each reward evolution step improves performance—directly contradicting the claim that no reward shaping was designed.
>
>
> ---
>
>
> ### **Response to W3: Evaluation Reliability and Human Study**
>
> See the **general response:** "1. Human evaluation results", "2. Alignment analysis between human evaluators and VLM evaluators", "3. Towards reliable and variance-aware evaluation with VLM evaluators".

---

> ### Author Response · Authors · 2025-11-26
> **Rebuttal (2/2)**
>
> ### **Response to W4: Comparison with Uicoder & ScreenAI**
>
> We thank the reviewer for suggesting *UIcoder* and *ScreenAI*. We agree that comparing with specialized UI-to-Code models is valuable. We have updated our paper to discuss them. However, we respectfully point out that **quantitative comparisons with these specific examples are infeasible due to distinct technical barriers**: *ScreenAI* is not open-sourced, and *UIcoder* relies on text/DOM inputs rather than the visual inputs used in our task.
>
> 1. **Task Mismatch (Text-to-Code vs. Vision-to-Code):** As the reviewer correctly noted, our work focuses on the Vision-to-Code paradigm (VLM), where the model must infer structure from raw pixels (screenshots). UIcoder [1] primarily operates as a Language Model (LM), taking text-based inputs rather than raw visual inputs.
> 2. **Reproducibility Constraints:** Regarding ScreenAI [2], while it shares the VLM setting, neither the model weights nor the inference code have been publicly released. This prevents us from performing a verifiable comparison. In our related work section, we also conducted an extensive search for specialized UI-to-code models, including ScreenAI (as suggested in , WebCode2M, Flame, and others mentioned in our Related Work. Unfortunately, none of these models have released their weights or inference code to the public. This lack of availability prevents us—and the broader community—from performing verifiable comparisons.
>
>
> ---
>
>
> ### **Response to W5: Supplementary ablation analysis (Impact of Pretraining vs SFT vs RL, Think/Answer, Comparator).**
>
> We conduct all supplementary ablation studies on Desigin2Code benchmarks.
>
> **1. Impact of Training Stages (Pretrain vs. SFT vs. RL)**
>
> Our Continual Pre-training (CPT) stage injects domain knowledge using datasets like WebSight and our crawled corpus (~10M pairs), establishing fundamental vision-code alignment capabilities. While the CPT model grasps HTML syntax and visual structures, it lacks instruction alignment. Consequently, when tested on Design2Code, it yields a low success rate (53.3%), often failing to adhere to formatting constraints or generating unstopped sequences despite producing syntactically correct HTML. As shown in the table, the SFT stage effectively bridges this gap, establishing a strong baseline (79.3%). Furthermore, our RL stage significantly boosts performance over this SFT baseline (improving from 79.3% $\to$ 88.6%), highlighting the value of visual feedback optimization.
>
> | Training Stage                 | Accuracy (%) | Absolute Gain | Primary Contribution                       |
> | ------------------------------ | ------------ | ------------- | ------------------------------------------ |
> | Base                           | 9.0          | -             | General Visual Recognition                 |
> | + Continual Pre-training       | 53.3         | **+44.3**     | HTML Syntax & Vision-Code Alignment        |
> | + Supervised Fine-Tuning (SFT) | 79.3         | **+26.0**     | Instruction Following & Format Constraints |
> | + Reinforcement Learning (RL)  | 88.6         | **+9.3**      | Pixel-level Visual Refinement              |
>
>
>
> **2. Impact of the "Think/Answer" Format**
>
> We validated the impact of the "Think/Answer" format by comparing it against a "Direct Answer" baseline. The result shows that the "Think/Answer" strategy yields a consistent improvement (+3.1% on Design2Code). The "Think" stage allows the model to perform a global structural analysis (e.g., identifying high-level regions like Header and Main_Content) before committing to implementation. This "visual planning" step ensures the generated code adheres to the correct DOM hierarchy, reducing layout hallucination.
>
> | Thinking mode | Design2Code |
> | ------------- | ----------- |
> | w/ thinking   | 88.6        |
> | w/o thinking  | 85.5        |
>
>
>
> **3. Impact of the Round-Robin Comparator**
>
> **We have already conducted the experiment in the original paper.** Table 3 (a) reports the superiority of the Round-Robin Comparator compared to other reward designs. Our proposed Comparator-based method (including the round-robin strategy) achieves 86.0%, which is a significant +3.0% improvement over the standard RL baseline (83.0%).

---

### Author Response · Authors · 2025-11-26
**General Response --- Towards Reliable and Comprehensive Evaluation (5/5)**

### **4. Cross-Verification with Additional Held-out Evaluators  (Reviewer 8XSC)**

Following the suggestion of Reviewer 8XSC, we further validate the soundness of our VLM-based judge by incorporating cross-verification with a diverse set of traditional and model-based metrics. For **element-matching metrics**, we include all metrics proposed in the original Design2Code[5] (Block, Text, Position, Color). For **vision feature–similarity metrics**, we employ both the CLIP (ViT-B-32) similarity used in *Design2Code* and the DINOv3 similarity (facebook/dinov3-vitl16-pretrain-lvd1689m) as suggested. All experiments are conducted on the *Design2Code* benchmark, and the results support the following conclusions:

1. **According to the held-out human evaluators, UI2Code^N w/ RL achieves performance comparable to state-of-the-art VLMs**, **which is consistent with the assessments provided by our VLM judge, confirming the validity of our reported performance.**
2. Across different models, all metrics yield **highly consistent rankings**, with GPT-5, Gemini-2.5-Pro, and UI2Code^N forming the top tier, while Qwen2.5-VL-72B consistently ranks the lowest among them.
3. **In terms of calibration, the VLM-based judge provides clearer and more discriminative performance gaps across methods, and the magnitude of these gaps aligns more closely with human evaluation.** This demonstrates that the VLM judge captures human-perceived quality differences more faithfully than traditional or vision-feature–based metrics, making it better suited for both guiding RL training and differentiating among top-performing models. This advantage stems from the VLM’s ability to perform deep visual–semantic analysis, which surpasses the limitations of embedding-based similarity metrics and HTML-attribute–based metrics (e.g., Text, Color) that do not directly assess the rendered UI.

|                      | **Block** | **Text** | **Position** | **Color** | **CLIP similarity** | **DINOv3 similarity** | **VLM-based (our judge)** |
| :------------------: | :-------: | :------: | :----------: | :-------: | :-----------------: | :-------------------: | :-----------------------: |
|        GPT-5         |   89.1    |   94.2   |     86.4     |   78.0    |        81.6         |         87.7          |           89.7            |
|    Gemini-2.5-Pro    |   89.1    |   93.5   |     85.5     |   71.4    |        80.9         |         85.6          |           89.5            |
|   Claude-4-Sonnet    |   88.7    |   93.2   |     84.6     |   72.0    |        80.5         |         85.5          |           81.2            |
|    Qwen2.5-VL-72B    |   86.6    |   91.6   |     76.8     |   67.8    |        77.8         |         74.4          |           41.9            |
| UI2Code^N (ours)-SFT |   86.8    |   91.5   |     81.7     |   69.7    |        79.0         |         78.8          |           79.3            |
| UI2Code^N (ours)-RL  |   88.7    |   93.1   |     83.8     |   72.6    |        80.5         |         86.1          |           88.6            |





## References

[1] Lee et al. (2023). *RLAIF vs. RLHF: Scaling reinforcement learning from human feedback with AI feedback*.

[2] Xu et al. (2023). *ImageReward: Learning and evaluating human preferences for text-to-image generation*. In Advances in Neural Information Processing Systems 36 (NeurIPS 2023).

[3] Wu et al. (2023) *Human Preference Score v2: A solid benchmark for evaluating human preferences of text-to-image synthesis*.

[4] Jin et al. (2025). *Omni-Reward: Towards generalist omni-modal reward modeling with free-form preferences*. arXiv.

[5] Se et al. (2024). Design2Code: Benchmarking Multimodal Code Generation for Automated Front-End Engineering

---

### Author Response · Authors · 2025-11-26
**General Response --- Towards Reliable and Comprehensive Evaluation (4/5)**

### **3. Towards Reliable, Variance-Aware Evaluation with VLM Evaluators (Reviewer 3Wur)**

To address the reviewer’s concern regarding variance-aware reporting and the credibility of our conclusions, we conduct a systematic stability analysis of the VLM-based evaluator. Specifically, we re-run the full Design2Code benchmark **five times**, including both (1) the UI generation process of each evaluated model and (2) the VLM-based evaluation process. Four representative models are included in this study: Qwen2.5-VL-72B, GPT-5, and both the SFT and RL variants of our proposed UI2Code^N.

The results are summarized in the following table. From the multi-round evaluation, we observe:

- **The evaluation pipeline is stable and variance-aware.** Across all five runs, the **ranking of all models remains identical**, demonstrating that our evaluation setup—comprising both the VLM evaluator and the benchmark—is highly consistent. The variance across runs is small for all models (with the only exception being Qwen2.5-VL-72B, whose generally lower performance naturally leads to greater fluctuation). This confirms that our reported numbers are not artifacts of random variation.
- **Our proposed method is consistently strong and reliably competitive.** **The final UI2Code^N-RL (9B) model achieves performance comparable to the significantly larger commercial model GPT-5**, and does so across all repeated runs with the **lowest standard deviation** among all evaluated models. This reliability further supports the superiority and robustness of our approach.

|     Model      | Round 1 | Round 2 | Round 3 | Round 4 | Round 5 | **Mean** | **Std** |
| :------------: | :-----: | :-----: | :-----: | :-----: | :-----: | :------: | :-----: |
| Qwen2.5-VL-72B |  41.9   |  41.6   |  39.5   |  40.8   |  38.9   |  40.54   |  1.31   |
|     GPT-5      |  89.7   |  89.3   |  89.5   |  90.2   |  90.6   |  89.86   |  0.53   |
| UI2Code^N-SFT  |  79.3   |  79.0   |  78.8   |  79.9   |  78.7   |  79.14   |  0.48   |
|  UI2Code^N-RL  |  88.6   |  88.5   |  88.2   |  87.7   |  88.4   |  88.28   |  0.36   |

---

### Author Response · Authors · 2025-11-26
**General Response --- Towards Reliable and Comprehensive Evaluation (3/5)**

### **2. Alignment Analysis between Human Evaluators and VLM Evaluators (Reviewer 3Wur, 8XSC)**

Regarding the concerns about the reliability and human alignment of our evaluator, we conducted **two complementary human–VLM agreement studies on the same dataset** (Design2Code-HARD, 80 samples):

1. **Model-Level Decision Alignment (pairwise win/tie/loss):** Do VLM judges reach *the same model comparison decisions* as humans?
2. **Sample-Level Score Calibration (per-sample scoring):** Do VLM scores correlate with humans, separate good/bad samples, and behave monotonically?

**TL;DR:**  Across *both* decision-level and score-level analyses performed on the same dataset with independent expert annotators, our evaluator demonstrates:

- Strong model-level alignment with human preferences (Pearson 0.93, Spearman 1.0)
- Consistent preference direction across all baselines
- Conservative behavior, not biased toward our model
- Score alignment with humans (Pearson 0.65)
- Large discriminative power (d = 1.3–1.8)
- Monotonic, interpretable calibration structure
- High binary reliability (98% recall of human-preferred outputs)

These properties collectively match or exceed widely-used multimodal evaluators in recent alignment research [1–4], showing that our evaluator is **reliable, human-aligned, and appropriate for UI-to-code evaluation**.

---

Here we illustrate the details.

**2.1 Model-level decision alignment: Human vs VLM produce the same win/lose structure**

**Setting.** For each of the 80 samples and for each baseline model (GPT-5, Gemini-2.5-Pro, Claude-4-Sonnet, Qwen2.5-VL-72B), human annotators and the VLM judge independently rated each model’s output. Using these ratings, we compute pairwise comparisons (“Ours vs baseline”) and derive **Win / Tie / Loss** counts.

To reduce annotator variance, we follow common practice and use the Human-Avg preference derived from two expert annotators.

**Strong human–VLM correlation on model-level decisions**

We compute **net-win margin = (Win − Loss) / N** for each baseline:

|    Baseline     | Human-Avg Margin | VLM Margin |
| :-------------: | :--------------: | :--------: |
|      GPT-5      |      28.80%      |   23.80%   |
| Gemini-2.5-Pro  |      17.40%      |   1.20%    |
| Claude-4-Sonnet |      43.50%      |   45.00%   |
| Qwen2.5-VL-72B  |      95.00%      |   87.50%   |

**Correlations:**

- **Pearson(Human-Avg, VLM) = 0.93**
- **Spearman(Human-Avg, VLM) = 1.0**

The evaluator reproduces the **exact same ranking** of baseline difficulty (Qwen ≪ Claude < GPT-5 < Gemini), and matches human preference strength with nearly perfect monotonic alignment.

We further visualize the correlation between Human-Avg and VLM judge on pairwise margins across baselines at https://postimg.cc/G4Q9BzrB .

**Consistent preference direction & conservative behavior**

- For all baselines, human and VLM agree on the direction: Ours > baseline.
- The VLM is consistently more conservative than humans, especially on the strongest baseline (Gemini), demonstrating no evidence of evaluator favoritism.

These results directly address concerns about reward gaming, evaluator bias, and lack of cross-evaluator consistency.



---

**2.2 Sample-level score calibration: Human–VLM alignment**

**Setting.** Using the same 80 samples, humans assigned 0–5 quality scores to our model’s outputs (3 annotators × 80 samples), while the evaluator produced 0–100scores. This experiment reveals fine-grained score alignment, variance, and calibration structure, complementing the model-level decision study.

**Correlation with human perception**

- Per-sample human-score std = 0.233, indicating meaningful subjectivity. → Using mean human score as ground truth is necessary and standard.
- Pearson(VLM, Human-Mean) = 0.65
- Spearman(VLM, Human-Mean) = 0.44

These correlation magnitudes fall squarely within the range reported by widely adopted evaluators for subjective generative tasks—such as RLAIF-based evaluators [1], ImageReward [2], HPS v2 [3], and Omni-Reward [4]—which typically operate with human–model correlations in the 0.4–0.7 band.

**Large human–VLM separation between good vs bad outputs**

Using VLM ≥ 80 as a community-standard threshold:

- Human(VLM ≥ 80): 3.80–3.81
- Human(VLM < 80): 2.50–2.80
- Cohen’s d = 1.32–1.82 (extremely large)

This reflects over one full point difference on the human scale, showing strong discriminative power.


**Binary agreement: accurate and high-recall**

Using human ≥4 as “good”:

- Agreement: 82.3%
- Recall: 98%
- Precision: ~0.82
- Cohen’s κ = 0.41

The evaluator rarely rejects samples humans consider good (high recall), a desirable property for both evaluation and reward modeling.

---

### Author Response · Authors · 2025-11-26
**General Response --- Towards Reliable and Comprehensive Evaluation (2/5)**

### **1. Human Evaluation Results (Reviewer 3Wur, q7LE, 8XSC)**

**1.1 Human evaluation on UI-to-Code**

We evaluate UI-to-code performance on the **Design2Code-HARD** dataset, as recommended by Reviewer 8XSC. This benchmark contains 80 challenging samples specifically constructed to expose distinctions among state-of-the-art (SOTA) methods. To ensure a stable and fair comparison, we recruit two independent human annotators, each tasked with evaluating every sample generated by UI2Code^N (ours) against outputs from both closed- and open-source SOTA baselines—including Gemini-2.5-Pro, GPT-5, Claude-4-Sonnet, and Qwen2.5-VL-72B. Annotators assess multiple dimensions of quality, including visual structure and alignment, color and aesthetic design, and textual and content consistency.

Human (Average)：

|                          |  **Win**  | **Tie** | **Lose** | **Win+Tie** |
| :----------------------: | :-------: | :-----: | :------: | :---------: |
|      Ours vs. GPT-5      | 56.3% |  16.3%  |  27.5%   |    72.5%    |
| Ours vs. Gemini-2.5-Pro  | 52.5% |  12.5%  |  35.0%   |    65.0%    |
| Ours vs. Claude-4-Sonnet | 63.8% |  16.3%  |  20.0%   |    80.0%    |
| Ours vs. Qwen2.5-VL-72B  | 96.3% |  2.5%   |   1.3%   |    98.8%    |




For comparison, we provide the VLM judge results by comparing the individual scores given by the VLM judge on each sample:

|                          |  **Win**  | **Tie** | **Lose** | **Win+Tie** |
| :----------------------: | :-------: | :-----: | :------: | :---------: |
|      Ours vs. GPT-5      | 53.8% |  16.3%  |  30.0%   |    70.0%    |
| Ours vs. Gemini-2.5-Pro  | 40.0% |  21.3%  |  38.8%   |    63.8%    |
| Ours vs. Claude-4-Sonnet | 63.8% |  17.5%  |  18.8%   |    81.3%    |
| Ours vs. Qwen2.5-VL-72B  | 91.3% |  5.0%   |   3.8%   |    96.3%    |

**Results:**

- As visualized in https://postimg.cc/30d21V8q, **UI2Code^N (ours), our 9B open-source VLM, outperforms state-of-the-art open-source and commercial models (at the time of submission) including GPT-5, Gemini-2.5-Pro, Claude-4-Sonent, and Qwen2.5-VL-72B.**
- **The evaluation results between human judge and VLM judge maintain a very strong correlation**, which will be further analyzed at the second part.

---


**1.2 Human evaluation on UI-Editing**

To complement the qualitative UI-editing examples presented in the Appendix, we additionally provide quantitative results. We constructed a new benchmark, UIEdit-Bench, comprising 69 samples spanning diverse UI designs and themes, each paired with a corresponding editing instruction covering a wide range of operations such as layout adjustments and the addition or removal of specific components. Human annotators then evaluate the resulting UI-editing performance on a 0–5 scale, assessing (1) whether the requested edits were performed correctly, (2) whether the parts not intended to be edited were successfully preserved, and (3) the overall quality of the edit.

**Human evaluation results (see table below) show that our RL-enhanced model achieves the highest scores across all criteria, including edit correctness, preservation of unedited content, and overall quality.** It surpasses all commercial and open-source baselines, establishing a new state of the art in human-judged UI-editing performance.

|    **Model**    | **Correctness of Requested Edits** | **Preservation of Unedited Sections** | **overall** |
| :-------------: | :--------------------------------: | :-----------------------------------: | :---------: |
| Claude-4-Sonnet |                4.83                |                 4.54                  |    4.69     |
| Gemini-2.5-Pro  |                4.42                |                 4.17                  |    4.30     |
|      GPT-5      |                4.63                |                 4.46                  |    4.54     |
| Qwen-2.5-VL-72B |                3.53                |                 3.30                  |    3.41     |
|    ours-SFT     |                4.64                |                 4.57                  |    4.60     |
|     ours-RL     |              **4.94**              |               **4.80**                |  **4.87**   |

---

### Author Response · Authors · 2025-11-26
**General Response --- Towards Reliable and Comprehensive Evaluation (1/5)**

Thanks for the reviewers’ thoughtful suggestions regarding the comprehensiveness of our evaluation protocol. While UI-to-code is primarily an objective reconstruction task with clear ground-truth references, we acknowledge that concerns about evaluator reliability are reasonable, given our earlier reliance on VLM-based judges.

To address these concerns in a rigorous and multi-perspective manner, we extended our evaluation to complement and validate the original automatic evaluation protocol through the following layered and mutually reinforcing components:

1. **Human evaluation with multiple independent annotators**, providing objective assessments of UI-to-code fidelity and UI-editing quality. The results confirm that UI2Code^N achieves state-of-the-art performance that is consistent with the findings in the original paper.
2. **Alignment analysis between human evaluators and VLM evaluators**, examining both model-level ranking agreement and sample-level score correlation. The strong alignment shows that VLM judges are reliable for this reconstruction task and do not introduce systematic bias.
3. **Variance-aware evaluation with VLM evaluators.** We further assess the stability and reliability of VLM-based scoring through variance-aware evaluation, demonstrating that the evaluator yields consistent rankings and tightly bounded variance across repeated runs.
4. **Cross-verification with additional held-out evaluators**, including DINO v3 similarity, CLIP similarity, traditional DOM-element-based metrics, etc. In addition to consistent ranking patterns, we find that the VLM judge offers better calibration and stronger discriminative power than traditional metrics, providing a more faithful measure of semantic and structural fidelity.

Overall, our analyses show that the **VLM-based evaluator is reliable, stable, and closely aligned with human judgment**. **Under human evaluation and multiple held-out metrics, UI2Code^N consistently delivers state-of-the-art performance, matching or surpassing much larger commercial systems despite its 9B scale**. These results validate our paradigm of treating UI-to-code as iterative reasoning and self-correction with visual feedback, supported by improved reward design. **We further visualize more demo cases on anonymous webpage https://fabulous-syrniki-54b5f8.netlify.app/ to demonstrate our model's capability.**

As the first open-source VLM covering UI-to-code, UI polishing, and UI editing, with a full training recipe from pre-training to RL, we believe this work provides a practical and forward-looking foundation for UI-centric coding agents and, by aligning closely with real human UI workflows, offers a principled and reproducible direction that can help structure progress in this rapidly developing area.

---

### Meta-Review · Area_Chair_8ujJ · 2025-12-30

**Summary:**

This submission proposes an Interactive UI-to-Code paradigm that reframes UI-to-code as an iterative workflow with three unified capabilities: (i) UI-to-code draft generation, (ii) UI polishing via rendered visual feedback, and (iii) instruction-based UI editing. The paper introduces UI2Code$^N$, a 9B VLM trained via a three-stage pipeline (continual pretraining on large-scale crawled webpage pairs, supervised fine-tuning for the three tasks, and RL with a VLM-based verifier reward). Reported results are strong on standard UI-to-code benchmarks and on newly introduced polishing/real-world benchmarks, with an additional test-time scaling story through multi-round polishing.

Across the three reviews, there is broad agreement that the problem is important and that the interactive framing is intuitive and practically motivated. The primary factors driving disagreement are that two reviewers issued strong rejects mainly due to: (i) soundness concerns about evaluation (heavy reliance on VLM judges for both training reward and final evaluation; lack of human validation and variance-aware reporting), (ii) methodological transparency / attribution (insufficiently clear what drives gains without ablations), (iii) baseline coverage (lack of comparisons to certain relevant UI-to-code or agent-style systems), and ethics/data governance (large-scale web crawling, licensing/ToS/PII handling, and scalability/cost concerns).

The rebuttal substantially strengthens the submission, and the technical work appears promising. However, after incorporating expected post-rebuttal score movements, the paper would still likely have at least one reviewer below the acceptance threshold (and potentially two), driven by remaining novelty/positioning concerns and reproducibility/benchmark availability limitations, plus ethics/legal concerns around large-scale crawling that cannot be externally verified.

**Reviewer Concerns:**

**Concerns substantially addressed in the rebuttal**

(1) Evaluation reliability / “VLM judge only” critique (Reviewers 3Wur, q7LE, 8XSC)

The authors added multiple layers of validation: (i) Human evaluation on Design2Code-HARD (80 samples) with two annotators and pairwise comparisons vs GPT-5, Gemini-2.5-Pro, Claude-4-Sonnet, and Qwen2.5-VL-72B, reporting win/tie/lose rates. (ii) Human–VLM alignment analyses, including model-level ranking agreement (reported Pearson 0.93, Spearman 1.0 on net-win margins) and per-sample score correlations (Pearson 0.65). (iii) Variance-aware evaluation with repeated runs (five reruns reported) showing stable model ranking and small standard deviations for top systems. (iv) Cross-verification with additional non-VLM metrics (Design2Code-style block/text/position/color metrics, CLIP similarity, DINOv3 similarity) showing consistent rankings and improvements from SFT→RL.

(2) Missing ablations / attribution of gains (Reviewers 3Wur, 8XSC)

The authors added ablations that isolate: (i) Training stage contributions (continual pretraining → SFT → RL), including an explicit “SFT w/ pretrain vs w/o pretrain” comparison (reported 72.1 vs 63.6 on Design2Code in the rebuttal) and SFT→RL gain (reported +9.3 on Design2Code). (ii) Think/answer output format (reported +3.1 on Design2Code). (iii) Round-robin comparator reward improvements (they also point out ablations were already present in the original paper; the rebuttal restates the effect size).

(3) Baseline coverage: agent-based systems (Reviewer q7LE; also partially relevant to 3Wur/8XSC)

The authors added direct comparisons on Design2Code-HARD vs ScreenCoder and DCGen, including both quality metrics and a basic efficiency breakdown (latency/token cost). This meaningfully strengthens the baseline story, at least along the agentic axis.

(4) UI editing evaluation (Reviewers q7LE, 8XSC)

The authors introduced a UIEdit-Bench (69 samples) with human 0–5 scoring and report their RL model outperforming baselines on edit correctness, preservation, and overall quality. This addresses the “editing is only demos” critique.

(5) Reward model training details (Reviewer 8XSC)

The authors disclosed additional details for the polishing reward verifier, including: A distillation-style dataset creation using Gemini-2.5-Pro as teacher labeling for UI triplets, filtering to ~8.5k, and SFT setup details. A held-out human agreement study where the SFT-tuned verifier reportedly improves relative judgment accuracy (62% → 94%).

(6) Evaluator choice (Reviewer 8XSC)

The authors justified using o4-mini for pairwise UI-to-code scoring and Gemini-2.5-Pro for triplet polishing comparisons, supported by a human agreement pilot and a sensitivity analysis across multiple judges (rank-order consistency reported).

**Concerns only partially addressed**

(1) Novelty / contribution positioning (primarily Reviewer 3Wur; partly 8XSC)

While the rebuttal convincingly argues that the implementation is careful and effective, it does not fully resolve the perception that: The overall approach is a standard pretrain → SFT → RL recipe plus an intuitive iterative refinement loop, and the “interactive” framing overlaps conceptually with existing iterative UI prototyping / refinement directions (now explicitly including Sketch2code) and with agent-based UI-to-code systems.

Note: *AC agrees with author's "Formal Complaint to AC for Reviewer 3Wur point 2," that this work does have merit and using a standard pipeline does not hurt the work's novelry. However, since this section is for "reviewer's concerns and how they may change," above is the AC's conjecture on reviewer's opinion.*

(2) Reproducibility of new benchmarks and training data

A key remaining issue is that several major claims lean on: New benchmarks (UI2Code-Real, UIPolish-bench, UIEdit-Bench), and A large-scale crawled corpus (~10M pairs) used for continual pretraining.

The authors state they do not plan to release any curated datasets due to privacy/copyright concerns. This is understandable, but it materially limits the independent verification of the benchmark contributions, and community ability to reproduce or stress-test the reported gains.

(3) Dependence on proprietary systems

Even after cross-verification, the pipeline still relies on closed models for parts of the workflow: Teacher labeling for the verifier (Gemini-2.5-Pro), Benchmark evaluation judges (o4-mini, Gemini-2.5-Pro), Some baselines (GPT-5/Claude-4). This is increasingly common, but when combined with unreleased datasets, it amplifies reproducibility concerns and poses a higher expectation on the conceptual contribution beyond the model (that expires so fast nowadays) and data.

(4) Ethics/legal compliance and privacy at scale (Reviewer 8XSC ethics flag)

The authors provide a governance description (robots.txt, ToS exclusions, PII filtering, no raw release), but the main remaining issue is external verifiability without dataset release or an auditable sampling process.

**Reviewer Scores:**

Reviewer 3Wur (initial: 2)

Many concrete critiques (human eval, evaluator reliability, missing ablations, missing methodological detail) are meaningfully addressed. However, the reviewer’s core skepticism about contribution/novelty and lack of certain specialized comparisons is unlikely to fully disappear. (Again, AC does not find novelty to be a major concern.)

Expected change: 2 → 4

Reviewer q7LE (initial: 6)

The rebuttal directly answers all requested additions: agent-based comparisons, human evaluation, and a formal UI editing evaluation.

Expected change: 6 → 6

Reviewer 8XSC (initial: 2)

The rebuttal is comprehensive: human alignment, sensitivity checks across evaluators, held-out metric cross-verification, reward model training disclosure, pretraining ablation, oscillation analysis, and UI editing evaluation. Remaining issues likely center on ethics/legal verifiability and reproducibility (non-release of datasets), which may prevent a full flip to accept.

Expected change: 2 → 4, small change of 6.

---

### Decision · Program_Chairs · 2026-01-26

Reject